



# Meteorological and climatological triggers of past and present bark beetle outbreaks in the Czech Republic

Rudolf Brázdil[1,2], Petr Zahradník[3], Péter Szabó[4,5], Kateřina Chromá[2], Petr Dobrovolný[1,2], Lukáš Dolák[1,2], Miroslav Trnka[2,6], Jan Řehoř[1,2], Silvie Suchánková[4]

[1]Department of Geography, Faculty of Science, Masaryk University, Brno, Czech Republic
[2]Global Change Research Institute, Czech Academy of Sciences, Brno, Czech Republic
[3]Research Institute of Forestry Economics and Gamekeeping, Jíloviště, Czech Republic
[4]Department of Vegetation Ecology, Institute of Botany, Czech Academy of Sciences, Brno, Czech Republic
[5]Department of Environmental Studies, Faculty of Social Studies, Masaryk University, Brno, Czech Republic
[6]Department of Agrosystems and Bioclimatology, Mendel University in Brno, Brno, Czech Republic

*Correspondence to*: Rudolf Brázdil (brazdil@sci.muni.cz)

**Abstract.** Based on documentary evidence, a chronology of bark beetle outbreaks in the Czech Republic from 1781 to 1963 CE was created, continuing from 1964 through 2021 by bark beetle salvage felling data. The spatial distribution of bark beetle events concentrates on the border mountains of Bohemia and in the northern parts of Moravia and Silesia. The

temporal distribution of the most important bark beetle outbreaks concentrates in the 1830s, 1870s, 1940s–1950s, 1980s, 1990s, 2000s, and 2010s. Each of these calamities was analysed in detail with respect to their spatial extent, the volume of damaged wood, and their meteorological patterns. While meteorological triggers of the largest outbreaks of the 19th century were attributed especially to the slow procession of disastrous volumes of damaged wood after large windstorm events sometimes intensified by dying trees in subsequent dry years, the recent warming moves the main meteorological and

climatological triggers to more frequent warm and dry meteorological patterns from the 1980s, sometimes enhanced by uprooted and broken trees after heavy windstorms. The last bark beetle outbreak from 2015 was evaluated as the most disastrous disturbance to spruce forest over the territory of the Czech Republic in documented history. The paper also discusses uncertainties in bark beetle data, responses to past bark beetle events, and relationships between environment, climate, and bark beetle outbreaks.

## 25   1 Introduction

The recent warming with relatively stable or even decreasing precipitation totals was accompanied by important drought episodes during the 2010s on the central European scale, reflected in the appearance of severe meteorological, agricultural, and hydrological droughts with various negative impacts (Zahradníček et al., 2015; Hoy et al., 2017, 2020; Ionita et al., 2017; Zscheischler and Fischer, 2020). Among them bark beetle overpopulation (especially *Ips typographus*) has to be

mentioned, which caused a devastating calamity in the Norway spruce forests (e.g., Hlásny et al., 2021a, 2021b; Jaime et al., 2022). The reason is that dry episodes with high temperatures in the vegetation period decrease the vitality and structure of





spruce with shallow roots, which is reflected in less resistance to bark beetle attacks. On the other hand, droughts with high temperatures accelerate bark beetle evolution and increase its number of generations to three in lower altitudes and to two in higher ones (e.g., Pfeffer and Skuhravý, 1995; Jönsson et al., 2009, 2011; Netherer et al., 2019; Hlásny et al., 2021a).

Besides dry and warm patterns, other meteorological disturbances also play an important role. Especially after severe windstorms with large amounts of uprooted or broken trees, bark beetles use fallen, freshly dead trees – if such wood was not processed and removed from the forest stands relatively fast – as bases on which they propagate prodigiously and then invade healthy individuals, significantly increasing the original damage (e.g., Temperli et al., 2013; Thom et al., 2013; Stadelmann et al., 2014).

Bark beetle outbreaks have also become a serious problem for spruce forests on the European scale. For example, Schelhaas et al. (2003), quantitatively analysing the role of natural disturbances in European forests from 1950–2000, attributed 16 % of the related mean annual damage (35 million $m^3$ of wood) to biotic factors, half of which was caused by bark beetles. The most comprehensive recent study of the state of knowledge on drivers and impacts of bark beetle outbreaks in Europe comes from Hlásny et al. (2021a) with an extensive list of references. Among different aspects of bark beetle outbreaks, great

attention concentrates on their meteorological triggers and the potential effects of recent and future climate change. For example, Annila (1969) and Wermelinger and Seifert (1998, 1999) analysed in great detail the effect of temperatures on the duration of the egg, larval, and pupal stages of the spruce bark beetle and adult maturation feeding. Jönsson et al. (2009) used a model to analyse the development and swarming activity of bark beetles for three future climate scenarios for 1961– 2100 in Sweden. In a further paper, Jönsson et al. (2011) modelled the potential warming impact on bark beetle voltinism

and reproductive diapause in Scandinavia. Temperli et al. (2013) examined in three case studies for Germany and Switzerland the mechanisms and feedback driving short-term and long-term interactions among beetle disturbances, climate change, and windstorms. Thom et al. (2013) analysed spatiotemporal variation in disturbance damage in Austria, with particular attention to wind and bark beetle impacts. Stadelmann et al. (2014) investigated spatial interactions between storm damage and subsequent bark beetle infestations in central Switzerland. Marini

et al. (2017) concentrated on the climate drivers of the dynamics of bark beetle outbreaks in Norway spruce forests across eight European countries over the last three decades. Seidl and Rammer (2017) pointed out the importance of interactions between wind and bark beetle disturbances in Austria using data between 2007 and 2014, amplified by climate change. Matthews et al. (2018) demonstrated that tree transpiration deficit, depending on soil water limitation, increases host susceptibility to bark beetle attack. Jakoby et al. (2019) developed a model of bark beetle phenology and applied it to three

regional climate change scenarios in the diverse topography of Switzerland. Kunca et al. (2019) identified wind and bark beetles on Norway spruce as the two most damaging agents in forest stands in Slovakia based on salvage felling in 1998– 2017. Netherer et al. (2019) analysed the effect of acute drought as an important driver of bark beetle infestation in the forest stands in Austria. Sommerfeld et al. (2020) studied possible future occurrences of bark beetle outbreaks in the Bavarian



Forest National Park under different scenarios of climate change. Jaime et al. (2022), based on data from 2010 to 2018,

suggest that the influence of droughts and bark beetle outbreaks will threaten the persistence of European coniferous forests.

Information about bark beetle occurrences was a part of Czech forestry literature for a long time (e.g. Chadt-Ševětínský, 1913; Nožička, 1957; Hošek, 1981), but it also appears in many specialized papers. For example, Záloha (1970) reported a bark beetle calamity in consequence of the disastrous windstorm of 26–27 October 1870. Jelínek (1988) in a detailed study analysed the same devastating bark beetle outbreak of the 1870s in the Šumava Mts. Skuhravý and Šrot (1988) described the

bark beetle outbreak in 1982–1986. Mrkva (1993) concentrated on the drought of 1992 and the onset of the subsequent bark beetle calamity. Zatloukal (1998) characterised historical and current factors of the bark beetle calamity in the Šumava National Park. Skuhravý (2002) presented an overview of some bark beetle calamities in the Czech Republic within the European context. Hlásny et al. (2011) used the PHENIPS model to analyse the impacts of future climate change on the distribution and voltinism of spruce bark beetles in the Czech Republic. Lubojacký (2012) gave an overview of Czech

legislation related to protection against bark beetles. Berec et al. (2013) investigated bark beetle dynamics using the PHENIPS model to assess the impacts of climate change on the Bohemian Forest. Zahradník and Zahradníková (2019) analysed salvage felling caused by different abiotic and biotic factors in the Czech Republic for 1998–2017, including those caused by the bark beetle. Hlásny et al. (2021b) concentrated on the drivers, impacts, and management implications of the last devastating outbreak of bark beetle in the Czech Republic.

An unprecedented bark beetle outbreak began in recent years in the Czech Republic (Zahradník and Zahradníková, 2019; Hlásny et al., 2021b), starting from about 2003. Two questions arise. First, does recent climate change completely alter the conditions of such events? And second, how does this outbreak appear in the context of preceding bark beetle outbreaks? We must then analyse not only the conditions of more recent events, but similar situations from the past. This is the aim of the recent article, covering the period from the 18th through the 21st centuries. In order to fulfil this request, the most

comprehensive and unique series of bark beetle outbreaks in the Czech Republic was created for this study and used especially for the analysis of meteorological and climatological conditions accompanying these outbreaks.

Sect. 2, which follows, describes bark beetle and meteorological/climatological data, and Sect. 3 describes the methods used in this study. The results in Sect. 4 present a long-term bark beetle chronology and analyse the meteorological and climatological triggers of the most outstanding past bark beetle outbreaks. Sect. 5 discusses obtained results with respect to

data uncertainty, responses to past bark beetle outbreaks, and relationships among environment, climate, and bark beetle outbreaks. Sect. 6 summarises some conclusions.

## 2 Data

### 2.1 Bark beetle data

Different data sources were used to create a bark beetle database for the territory of the former Czech Lands, i.e. now the

Czech Republic (further CR). Much bark beetle information was obtained from a large project called "Forest History



Research" (*Historický průzkum lesů*). It was initiated in the early 1950s by *Lesprojekt* in Brandýs nad Labem (recently the Forest Management Institute, *Ústav pro hospodářskou úpravu lesů*) (for locations of places and regions in the CR see Fig. S1). A collection of detailed historical information about each forest district was contributed for the economic improvement of forests (Hošek, 1983). In total, 544 unpublished volumes from the work continuing until the late 1980s are now kept in the

archive of the Forest Management Institute in Brandýs nad Labem (Novotný, 2011). An example of the bark beetle information in these volumes is a report of a forester Franz Loschan from the Telč domain who recorded dry spruce trees invaded by bark beetle in the forest district Rosičky in the autumn of 1807 (unpublished source US1, p. 65). The forest officer Vinzenz Tibl on 5 September 1808 from Červené Poříčí wrote that "the bark beetle occurred in all forest districts, especially on the line Dubí in the Doupov district (a group of 16 dry trees and 45 heavily invaded) and on the line Horní

Stropecko in the Kbely district (12 dry trunks and 30 invaded in the surroundings)" (US2, p. 76). At the Loket domain, the forest stage manager Jan Wildt informed in a letter to a burgomaster office on 24 July 1870 that "many trees are invaded by the bark beetle, partly dry, in forest districts Kozí hřbet and Kovářská" (US3, p. 77). Bark beetle occurrences were often mentioned in connection with other natural disturbances like windstorms, snowstorms, droughts, etc. Reports also included various mitigation measures to prevent possible future bark beetle outbreaks. For example, after an 1870 windstorm, the

purchase of new horses to remove the windthrow wood from forests, a prohibition on logging healthy trees or selling the redundant trunks to merchants for the next processing were applied in the Prášily domain (US4, p. 68).

Annual forest reports with information on forest pests were published for a few estates (domains) in "Neue Schriften" of the "Imperial Royal Patriotic-Economic Society of Bohemia", together with meteorological and phenological observations (Brázdil et al., 2011) as well as agricultural reports for the years 1828–1846 (Neue Schriften, 1830–1847). For example, in

1833 on the Plasy estate, there was a bark beetle outbreak "which, aided by the weather, caused great damage." The local foresters tried to contain the outbreak by laying out spruce trunks without the bark to attract the beetles. In this way they destroyed more than a million bark beetles (Neue Schriften, 1836, p. 217).

Among other documentary sources must be mentioned information from professional papers in the form of monographs (e.g. Nožička, 1957; Hošek, 1981; Jelínek, 1988; Skuhravý, 2002) and from many individual papers published especially in the

*Lesnická práce* journal (e.g., Kalandra et al., 1957; Skuhravý and Šrot, 1988; Simanov, 2014) or in other journals (e.g., Kudela, 1980; Zatloukal, 2003). Further bark beetle information has also been found in newspapers (e.g., *Rudé Právo*, 29 June 1966, p. 4; ibid., 31 August 1983, p. 2), or via internet sources (e.g., Historie lesních kalamit v ČR, 2018).

Systematic quantitative data related to bark beetle forest damage for the 1964–2021 period was compiled by the Research Institute of Forestry Economics and Gamekeeping (*Výzkumný ústav lesního hospodářství a myslivosti*) at Jíloviště-Strnady in

the form of annual volumes of salvage felling attributed to bark beetle impacts. This data was collected from reports of individual forest districts (foresters) in the CR.



**2.2 Meteorological and climatological data**

Because meteorological and climatological factors influencing the evolution of bark beetle outbreaks are important triggers of their calamities in the CR, the following climatological data for the 1781–2021 period were used in this study:

(i) mean monthly areal temperature series for the CR (Brázdil et al., 2012), extended to 2021

(ii) mean monthly areal precipitation series for the CR (Brázdil et al., 2012), extended to 2021 and completed with seasonal and annual reconstructed precipitation totals before 1804 (Dobrovolný et al., 2015)

(iii) mean monthly areal self-calibrated Palmer Drought Severity Index (scPDSI) series for the CR (Brázdil et al., 2022a)

(iv) the chronology of windstorms in the CR based on data from the Historical-Climatological Database of the Institute of

Geography, Masaryk University at Brno.

**3 Methods**

Different documentary data of bark beetle occurrences (see Sect. 2.1) were critically evaluated to create a database containing information about the time (year), place, special bark beetle information (occurrence, type of damage, damaged volume of wood, measures against bark beetle), and source of report. Information about place was divided according to

ownership of forest stands such as estates (domains) or towns, sometimes specified to the corresponding forest district or even individual forest stand. Because some older quantitative data about the felling of wood invaded by bark beetle as well as the area of such affected stands were expressed in older units, they were recalculated to the recent units as follows: the volume unit 1 fathom (*sáh*) equals 2.842 m³ of stacked wood or 1.8954 m³ of solid wood (i.e., without the air between the logs); the areal unit 1 square measure (*jitro*) equals to 0.5754 ha.

The created database was used to describe long-term spatiotemporal variability of bark beetle occurrences over the territory of the CR between 1781 and 1963 CE, from which the three most outstanding bark beetle outbreaks based on the territorial extent and estimated volume of damaged wood were selected for their detailed study. These selected outbreaks were further complemented by four such events selected from the highest volumes of bark beetle salvage felling from 1964. The spatial distribution of bark beetle occurrences during the two outbreaks from the 19th century is characterised by estates (domains)

and localities with invaded forest stands, while selected outbreaks after 1963 were spatially attributed to the newly established 77 administrative districts (*okres*), expressing total volumes of bark beetle salvage felling for them as well as volumes in the individual years of a given outbreak. For the description of meteorological and climatological patterns, annual and summer (June–August) anomalies of air temperature, precipitation totals and scPDSI with respect to the 1961–1990 reference period were expressed. This reference was preferred compared to the more recent 1991–2020 period because

of more stable climatic patterns (Brázdil et al., 2022b). For corresponding monthly anomalies, their cumulative sums were calculated starting from the January anomaly of the first year of the related bark beetle outbreak, and finishing by the addition of the December anomaly of the last year. A description of climatological patterns was complemented by the



reporting of windstorms as a meteorological disturbance, taken from our historical climatological database. The events from 1964 were further complemented by annual volumes of salvage felling, separately attributed to bark beetle and windstorms.

In order to characterise general climatological conditions of all seven selected bark beetle outbreaks together, a composite or superposed epoch analysis was applied (Haurwitz and Brier, 1981), calculating the mean summer and annual anomalies of temperatures, precipitation, and scPDSI five years before and five years after the first year of the corresponding outbreak (see Sect. 4.2.8). The statistical significance ($p < 0.05$) of these anomalies was estimated using a random bootstrapping approach (Rao et al., 2019).

To analyse large-scale circulation patterns corresponding to years around bark beetle outbreaks, we used an objective classification of circulation types based on flow strength, flow direction, and vorticity (Jenkinson and Collison, 1977; Plavcová and Kyselý, 2011), calculated with respect to the geographic midpoint of the CR from sea level pressure data in a NCEP/NCAR reanalysis (Kalnay et al., 1996). This classification defines nine anticyclonic types, nine cyclonic types, and eight directional types; the type U stands separately for unclassified patterns (see Řehoř et al., 2021a for more details).

Relative frequencies of circulation types in years around outbreaks were expressed as differences with respect to corresponding mean frequencies of the 1961–1990 reference period. These relative differences were further tested by the two-proportion Z-test (Sprinthall, 2011) for their statistical significance ($p < 0.05$).

## 4 Results

### 4.1 Bark beetle outbreaks during the 18th–21st centuries

The spatiotemporal chronology of bark beetle occurrences on the territory of the CR covers the period from the 1780s to the early 1960s CE. The beginning of this series is characterised only by a few reports before 1780 and a lack of spatial details after 1950, reporting rather larger areas like the borderland mountains. Fluctuations in the number of places with bark beetle occurrence follow from Fig. 1 and reported locations across the territory of the CR appear in Fig. S1. After 1963, bark beetle occurrences are presented generally for the whole CR according to corresponding volumes of salvage felling (see Fig. 2).

### 4.1.1 The 18th century

The first, but only general, reports of bark beetle occurrences on the territory of the CR already appear for the years 1720, 1733, and 1748 CE (Nožička, 1957). The number of related reports increased during the 1780s and 1790s. For example, in 1782–1785 bark beetle occurrence was documented especially in the western half of Bohemia (e.g., the Křivoklát, Loket, Plzeň, Toužim, and Zbiroh estates). After a windstorm on 5–6 November 1786, the invaded trees in the Jizerské hory Mts.,

the Krkonoše Mts., and the Orlické hory Mts. should have been processed for 15 years. Around 1790 a farm estate in Jindřichovice in the Krušné hory Mts. reported around 20,000 fathoms (56,840 m$^3$) of bark beetle wood. A further portion of bark beetle reports between 1793 and 1798 appear for several other estates having in their possession forest stands in Bohemia (e.g., Horšovský Týn, Kynžvart, Opočno, Přísečnice, Zbiroh). On 20 April 1799, the forester Ehrenwert reported to



*Gubernium* in Prague that bark beetles in the Loket region invaded 5,469 cadastral *jitro* (3,147 ha) of forest stands, from
which it was necessary to cut down 321,828 trees (US5, p. 93).

### 4.1.2 The 19th century

In the 19th century, the first bark beetle wave after some fallen trees in several meteorological disturbances came between
1807 and 1809, when the estates and regions of Bechyně, Červené Poříčí, Choceň, Jáchymov, Kumburk, Litoradlice, Plzeň,
Rokycany, or Slapy in Bohemia and Fulnek, Krnov, or Telč in Moravia and Silesia were affected. For example, 60,000 trees
were invaded in the Častolovice and Choceň forests (US6, p. 78). Further regions infested by bark beetle were reported in
1811–1813 for the Krušné hory Mts. and the estates Chrudim, Heřmanův Městec, Jičíněves-Veliš-Vokšice, Kumburk and
Třeboň.

An important expansion of bark beetles was recorded in the 1820s, when starting from 1822 its occurrence was reported in
each year of this decade for the different estates or localities of Chodová Planá, Jáchymov, Jezeří, Jindřichův Hradec,
Kardašova Řečice, Lovosice, Nepomyšl-Krásný Dvůr, Nové Hrady, Orlík, Ostrov, Přísečnice, Rokytnice, Rychmburk,
Toužim, or Třeboň in Bohemia, while in Moravia and Silesia it was Hranice, Libava, Lipník nad Bečvou, Potštát, or Žďár
nad Sázavou. For example, in Prášily forests in the Šumava Mts. area, 14,671 fathoms (41,695 m$^3$) of thrown and fallen good
trees and 5,000 fathoms (14,210 m$^3$) of trunks of over-aged trees appeared after the 20–25 November 1821 windstorm. To
avoid the expansion of the bark beetle population, a holder of a corresponding domain asked in August 1822 to speed up the
processing of the wood, even at the increased labour costs this would entail (US4, p. 67).

Windstorms in December 1833 and January 1834 followed by subsequent summer droughts in 1834–1835 were triggers of
the next and much larger bark beetle outbreak during the 1830s, which was documented in forest stands of 50 estates and 28
other localities (see Fig. 1 and Sect. 4.2.1 for more details).

The beginning of the 1840s signalled a subsiding of the preceding great bark beetle calamity with beetle occurrence only at a
few estates (e.g., Loket, Manětín-Rabštejn, Okrouhlice, Přísečnice, Toužim, Uhrov, Želiv, Žichovice), mainly in 1841 and
1843 (in the Písek region until 1845). For the next two decades, bark beetle occurrences were reported only for the estates of
Jezeří (1854), Branná (1861) and Bruntál (1865). But after two extreme windstorms in 1868 and 1870, the most disastrous
bark beetle outbreak of the 19th century occurred, continuing until 1875–1877. Our database documents this event in the
forest stands of 35 estates and 10 other localities, when especially the area of the Šumava Mts. was affected disastrously (see
Fig. 1 and Sect. 4.2.2 for more details).

After this bark beetle disaster, information about this pest occurrence was only sporadic for the following years up to the end
of the 19th century. For example, the increased occurrence of bark beetles around 1883 in the Vysoké Mýto region was
eliminated by peeling affected trees and by bark burning (US6, p. 240). In the forests of the Maršov domain, the bark beetle
was kept in harmless boundaries using catchers in 1888–1894, with the annual felling only between 55 and 167 m$^3$ of solid
wood (US7, p. 43).





### 4.1.3 The 1901–1963 period

Hošek (1981) estimated bark beetle calamities on the territory of the CR during 1900–1940 for 2.2 million m³ of wood. The first portion of bark beetle reports appeared after snow-fallen trees in 1904 and some blowdown especially in 1905–1908 for different estates in Bohemia like Ahníkov, Cvikov, Červený Hrádek, Debrné, Hořice, Hůzová, Jirkov, Náchod, Rokytnice,

Rychnov-Černíkovice, or Skály. In the following decade, the 1910s, only very sporadic bark beetle records occurred. In the 1920s, bark beetle occurrences were reported particularly after a disastrous calamity caused by nun moths (*Lymantria monacha*). To this is attributed damage in spruce and pine forests of *c*. 20 million m³ of wood across an area of 62,000 ha during the years 1917–1927 (Komárek, 1931; Simanov, 2014). But bark beetles did not achieve any broader or calamitous expansion. Only Kudela (1980) reported c. 22,000 m³ of bark beetle wood for 1922 in the surroundings of Nové Hrady and

České Velenice in southern Bohemia after neglecting blowdown trees from 1920. Skuhravý (2002) estimated the volume of damaged wood by nun moth and bark beetle in the mountain positions at 300,000 m³ of wood between 1922–1927.

As for the 1930s, greater expansions of bark beetles were reported for only some estates or localities in Bohemia (e.g., Boleboř, Přísečnice, Rychmburk, Rychnov-Černíkovice, Skály) and Moravia and Silesia (e.g., the areas around Velké Losiny and Velké Vrbno). A lack of forest workers during World War II (1939–1945) led to the slow processing of

blowdown trees in forest stands after several meteorological disturbances; in 1946 a bark beetle outbreak affected nearly all mountainous forest stands across the territory of the CR and their elimination continued until 1954 (see Sect. 4.2.3 for more details).

Further quantitative estimates from the 1950s include for 1953–1955 only 200,000 m³ of bark beetle wood in the border mountains and the Doupovské hory Mts. and 120,000 m³ for 1959–1960, again in the border mountains (Kudela, 1980).

Fewer volumes were reported in annual overviews of insect pests and diseases: 40,000 m³ in 1958 (Pivetz et al., 1959), 43,500 m³ in 1959 (Šrot et al., 1960), and 75,000 m³ in 1960 (Martinek and Šrot, 1961). Hošek (1981) estimated bark beetle salvage felling during 1961–1963 at *c*. 60,000 m³ of wood annually.

### 4.1.4 The 1964–2021 period

Systematic quantitative data of bark beetle salvage felling for the entire CR territory appears since 1964 with 287,000 m³ of

wood in that year. Corresponding volumes of such wood did not achieve a half-million m³ until 1982 (Fig. 2). But in 1983–1984 and 1986–1987 they overcame 1 million m³ annually. In 1993–1995 they were already above 1.5 million m³ per year, similar to 2008–2010. Then from 2012, with the value of 633,000 m³, a continuous increase in bark beetle salvage felling started and reached nearly 1.5 million m³ in 2015. Later during 2016–2017 it overcame 3 million m³ per year, in 2018, 8 million m³, and in 2019–2020 14 million m³ (to the maximum of 14.9 million m³ in 2020). In 2021 related salvage felling

declined to 9.5 million m³ of wood. Bark beetle calamities of the 1980s, 1990s, 2000s, and 2010s–2020s are analysed in detail in Sect. 4.2.4–4.2.7.





### 4.2 Meteorological and climatological triggers

Figure 3 demonstrates long-term fluctuations in the mean annual and summer series of air temperature, precipitation totals, and scPDSI over the territory of the CR during the 1781–2021 CE period. The mean annual temperatures slightly declined

from the warmer 1790s to *c*. 1890 while afterwards their continuous increasing tendency began, especially enhanced from the late 1980s (Fig. 3a). The mean annual precipitation totals after lower values before *c*. 1810 experienced a slightly decreasing tendency from the 1810s to the 1860s, and afterwards during an increase in the 1870s they kept higher totals, with the maximum of the entire series occurring around 1940 CE. Afterwards, totals kept a relatively similar level as before (Fig. 3b). Annual scPDSIs show similar behaviour as fluctuations in precipitation totals as high as the maximum (wet patterns)

around 1940, but afterwards a clear decreasing trend (i.e., increasing dryness) was well expressed, ending with an extremely sharp drop in the 2010s (Fig. 3c). This last, sharply enhanced dryness has no analogue in the entire scPDSI series. As for summer series, they slightly differ from annual series in individual values and in the amplitude of smoothed local extremes, but otherwise they express very similar temporal changes.

Because of the importance of meteorological and climatological triggers for the origin and course of bark beetle outbreaks,

the most outstanding bark beetle events in the whole chronology since 1781 CE with respect to their meteorological and climatological conditions in the CR are characterised in greater detail in the following sections.

### 4.2.1 The bark beetle outbreak of the 1830s

Locations of 78 estates and localities, whose forest stands were affected by the bark beetle outbreak in 1834–1839, are shown in Fig. 4a. Affected locations occur especially in southern, western, northwestern and eastern Bohemia, and in

northeastern Moravia and Silesia.

The meteorological trigger of this bark beetle outbreak was primarily a heavy windstorm that occurred on 18–19 December 1833, followed by high winds on 22 and 26 December and then by another windstorm on 31 December 1833–1 January 1834 (Brázdil et al., 2018b). Vicena (1964) estimated related damage to 4.2 million m³ of wood for the first windstorms and 0.3 million m³ for the second windstorm. Hošek (1981) reported the same volume of damaged wood for the first event,

divided into 3.7 million m³ in Bohemia and 0.5 million m³ in Moravia and Silesia (mainly in the Jeseníky Mts.). A large number of fallen and broken trees being processed together with a warm (especially summer) and dry year in 1834 and another dry year in 1835 (the lowest scPDSI) compared to the 1961–1990 reference (Fig. 4b) created good conditions for the bark beetle outbreak. For example, while the January windstorm fell 21,978 m³ of wood in the Vimperk domain, the subsequent bark beetle outbreak infested nearly ten times more (202,653 m³ of wood until 1839). In the forest of Volary

town, a lax approach to blowdown tree processing finally led to 151,603 m³ of damaged wood, from which more than half was attributed to bark beetle infestation (Jelínek, 1988). Zatloukal (2003) reported for the Šumava Mts. area *c*. 300,000 m³ of bark beetle spruce wood. Cumulative monthly precipitation anomalies showed a stable deficit between 250 and 300 mm from the summer of 1835 to the autumn of 1837, while decreasing cumulative monthly anomalies of scPDSI from late 1834

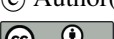



document increasing dryness which culminated in 1836, being continuously reduced afterwards by subsequent colder and
wetter years until 1839 (Fig. 4c).

### 4.2.2 The bark beetle outbreak of the 1870s

The location of 45 documented estates and localities over the territory of the CR, whose forest stands were affected by a bark
beetle outbreak in 1870–1875, is shown in Fig. 5a. Especially affected was the area of the Šumava Mts. in southwestern
Bohemia. For example, Jelínek (1988) reported that the Vimperk domain had bark beetle salvage felling in the volume of
1,079,115 m³ of wood during 1868/69–1877/78 decade. Further documented places were located in western Bohemia, in the
Krušné hory Mts., and the Krkonoše Mts. Interesting is the location of affected localities in a line from southern Bohemia to
the northeast, with further occurrences in the broader Jeseníky Mts. area (Fig. 5a).

The bark beetle outbreak in the 1870s had its meteorological triggers in two disastrous windstorms following shortly after
each other. The first windstorm on 7 December 1868 resulted in 6 million m³ of solid wood, according to Hošek (1981).
However, based on hard data from 273 forest districts throughout the Czech Lands, Brázdil et al. (2017) disclosed only 4.89
million m³ of solid wood. The subsequent disastrous windstorm on 26–27 October 1870 completely devastated many
forested areas, especially in the Šumava Mts. in southwestern Bohemia (Brázdil et al., 2018a). Hošek (1981) estimated total
forest damage during this windstorm at 4 million m³ of solid wood. A huge amount of uprooted and broken trees that could
not be processed relatively fast created optimal conditions for bark beetle spread during generally cooler years and more-or-
less normal summers, but also during drier years 1870–1871 (the lowest scPDSI already in 1870) compared to the 1961–
1990 reference (Fig. 5b). The cumulative monthly precipitation deficit was above 200 mm in 1872–1873 and increased by
more than 350 mm from autumn 1874 to summer 1875 (Fig. 5c). The dryness expressed by cumulative monthly scPDSI
anomalies was enhanced until 1874, while in the next year 1875 cumulative anomalies remained more or less constant.

The 1870 windstorm also damaged more than 620,000 m³ of wood on the German side of the southwestern border of the CR;
together with the 1868 windstorm and the subsequent bark beetle outbreak, 12.3% of all the forests were devastated there
(Elling et al., 1987). On the Czech side, salvage logging reached 3.5 million m³ of solid wood between 1868 and 1882
(Jelínek, 2005). Analysing the effects of the two windstorms and the following bark beetle outbreak for the Šumava Mts. and
Bayerischer Wald together, Brůna et al. (2013) estimated damage to *c*. 40 % of their 43,247 ha area, for which forests were
at least partly disturbed (7,725 ha totally, 4,647 ha by half). According to Svoboda et al. (2012), the effects of this disaster
remain still visible in many stands in the Šumava Mts.

### 4.2.3 The bark beetle outbreak of the 1940s–1950s

While post World War II bark beetle outbreaks affected forest stands in practically all border mountains of the CR,
information about volumes of damaged wood differs significantly. For example, Kalandra et al. (1957) reported *c*. 10 million
m³ of bark beetle wood during the first part of this outbreak in 1944–1947 and then in 1949–1954, emphasising the
weakening of spruce trees after a severe drought in 1947. Hošek (1981) mentioned 8 million m³ for the whole 1946–1954





period, giving annual felling data only for 1950–1954 in the total of 3.737 million m³ (1950 – *c*. 957,000 m³, 1951 – 720,000 m³, 1952 – 740,000 m³, 1953 – 800,000 m³, and 1954 – 520,000 m³). The bark beetle outbreak especially affected the border mountain systems of the CR as the Šumava Mts., the Krušné hory Mts., the Lužické hory Mts., the Jizerské hory Mts., the Krkonoše Mts., the Orlické hory Mts. and the Jeseníky Mts., but also the Doupovské hory Mts. in western Bohemia. Besides these regions, bark beetle occurrences were documented in excerpted documentary sources only for 11 different localities (Fig. 6a).


Kudela (1980) reported a snowthrow on 6–8 December 1939, especially in western, southern, and central Bohemia, a windstorm on 4 November 1941, and strong winds in two subsequent years as causes of *c*. 10 million m³ of broken and fallen trees in Czech forest stands, which were only slowly processed due to problems with a lack of forest workers, who moved during the war to the arms industry. This created good conditions for the bark beetle outbreak that appeared in full intensity during 1945–1947 with 2 million m³ of invaded wood. A further 0.2 million m³ of wood was reported for 1953–1955. While in 1952 trees threatened by bark beetle were cut down from the Novohradské hory Mts. over the Šumava Mts., the Český les Highland, the Doupovské hory Mts., and the Krušné hory Mts. to the Děčín region (Kalandra, 1953), small bark beetle pockets in the Ústí nad Labem region, the Jeseníky Mts., partly Moravskoslezské Beskydy Mts. and elsewhere were reported in 1955 (Kudler et al., 1956).



Compared to the 1961–1990 reference period, annual anomalies characterised the years 1945 and 1948–1951 as warmer (especially warm summers in 1947 and 1950), 1947 and 1953 as dry and 1947–1950 with the highest scPDSI negative anomalies (Fig. 6b). While cumulative monthly temperature anomalies were positive through the whole 1945–1954 period except for the first half of 1947, cumulative monthly precipitation anomalies were both positive and negative, achieving *c*. 100 mm in February–April 1947 and *c*. –150 mm in the first half of 1954 (Fig. 6c). In the case of cumulative monthly scPDSI anomalies, the continuously-increasing dryness appeared from the spring of 1947 to mid-1952, and did not change much afterwards until the end of 1954. Thanks to the wetter years of 1945–1946, cumulative values were not so strongly expressed as the known very dry year of 1947 (Brázdil et al., 2016).


Concerning this bark beetle outbreak, Skuhravý (2001) reported its occurrence in a broad Central European area extending from Switzerland and Austria to northern Germany on the one hand, and from the River Rhine in the west to southern Poland, Slovakia, and Hungary in the east on the other. Bark beetles invaded more than 30 million m³ of wood, especially in altitudes between 150 and 600 m a.s.l. This outbreak in southwestern Germany during 1944–1951, especially in Baden-Würtenberg and Rheinland-Pfalz, was analysed in the volume of papers edited by Wellenstein (1954). Turček (1950) described the situation in Slovakia during 1947–1948.


### 4.2.4 The bark beetle outbreak of the 1980s


The bark beetle salvage felling in 1982–1987 reached in the CR 6.205 million m³ of wood according to data in Fig. 2. The highest values (Fig. 7a) were recorded especially in the border districts of Bohemia, with most of those in northern Bohemia (Liberec district 611,604 m³), followed by three other districts with more than 250,000 m³ in southwestern Bohemia in the



Šumava Mts. (Klatovy 334,080 m³ and Prachatice 262,917 m³) and in northwestern Bohemia (Karlovy Vary 262,517 m³). In

Moravia and Silesia, the highest bark beetle felling reached 198,341 m³ in Šumperk district.

As follows from Figs. 7b and 7c, the warm and dry years 1982 and especially 1983 (1961–1990 reference period) led to a deficit in cumulative monthly precipitation anomalies achieving nearly 200 mm from late 1983 into the first half of 1985, which was reflected in continuously-increasing dryness expressed by scPDSI to the end of 1984. Afterwards, the situation started to improve due to cooler and wetter patterns in 1985 and 1987. As for summers, those in 1982 and 1983 showed a

positive anomaly of 1.1°C and 1.8°C respectively, while precipitation totals in the summer of 1983 were 58 mm below the long-term mean. But in this context, two extreme windstorms on 12–13 July 1984 and 23–24 November 1984 in the CR (Brázdil et al., 2018b) cannot be omitted which were significantly reflected in the important increase of windstorm salvage felling in 1984–1985 with 12.43 million m³ of wood together (Fig. 7b). Another severe windstorm on 19–20 January 1986 (Brázdil et al., 2018b) had less influence on bark beetle outbreaks that had significantly increased related salvage felling in

1986–1987 (8.28 million m³ together).

Skuhravý and Šrot (1988), who analysed in detail the same outbreak for 1982–1986, pointing out 5.081 million m³ of bark beetle wood, reported among meteorological triggers windstorm and snowstorm, dry episodes, and also air pollution. But they attributed this outbreak additionally to the neglect of cleaning and protecting spruce stands, to the underestimation of bark beetle danger in the early stage of its gradation (for example, the quantity of bark beetle increased between 1982 and

1983 3.5 times in the whole CR, but 12.9 times in the East-Bohemian region), and to the human factor.

### 4.2.5 The bark beetle outbreak of the 1990s

Taking into account bark beetle salvage felling in 1992–1995, the highest values were recorded especially in districts extending from southwestern Moravia to the northeastern part of the CR (Fig. 8a). Maximum volumes of bark beetle wood were felled in the Opava district with 513,382 m³ of wood, followed by the districts of Brno-venkov (386,059 m³), Bruntál

(384,318 m³), and Třebíč (364,999 m³), with three other districts with more than 200,000 m³. In Bohemia, the highest felling in Klatovy district reached only 120,377 m³. The total volume of bark beetle salvage felling in the CR during 1992–1995 reached 5.59 million m³ of wood (Fig. 2).

The dry and warm patterns of 1992 (Figs. 8b and 8c) with a particularly warm summer created good meteorological conditions for the subsequent bark beetle outbreak as warned by Mrkva (1993). Although the cumulative deficit of monthly

precipitation anomalies compared to the 1961–1990 reference was around 100 mm and more in some months of 1993–1994, increasing temperatures intensified the dryness significantly until the winter of 1994/1995. The situation slightly improved in the warmer, but also wet year of 1995, but without any important changes in the cumulative monthly anomalies of scPDSI. This bark beetle event remained outside of the potential impacts of two disastrous windstorms in 1990 (Vivian on 26 February and Wiebke on 1 March), because related wood from these windstorms was already processed during 1990–1991

in the total of 12.51 million m³ (Brázdil et al., 2018b), and corresponding volumes of salvage felling in 1992–1995 were



annually only *c*. 2 million m³ or less (Fig. 8b). On the other hand, in 1993–1995 and 1997, a significant increase in salvage felling attributed to drought (6.72 million m³ together) was recorded (not shown).

### 4.2.6 The bark beetle outbreak of the 2000s

A herald of this bark beetle outbreak in the CR was already related to salvage felling in 2003–2004 with a total of 1.94 million m³, triggered by a very warm and dry weather in the summer of 2003. Although some papers (e.g. Hlásny et al., 2021b) took this as a bark beetle event continuing until recent times, we are dividing it into two episodes with significantly increased bark beetle salvage felling, namely in 2007–2010 and after 2015. The first outbreak of 2007–2010 (Fig. 9a) was concentrated especially in the Šumava Mts. in southwestern Bohemia (the district Klatovy with 796,706 m³ of wood and Prachatice with 611,530 m³) and in the districts in the northern part of Moravia and Silesia (Bruntál 498,726 m³ and Opava 390 483,959 m³). The salvage felling of over 200,000 m³ of wood was also recorded in three other districts in southern (Český Krumlov) and eastern (Chrudim) Bohemia, and in northern Moravia (Frýdek-Místek). The total volume of bark beetle salvage felling in the whole CR during 2007–2010 reached 6.66 million m³ of wood (Fig. 2).

While annual mean temperatures experienced high positive anomalies in 2007–2010 compared to the 1961–1990 reference, reflected in the continuous increase of cumulative monthly temperature anomalies in 2007–2009, cumulative monthly 395 precipitation totals fluctuated only between –50 and 50 mm during these three years, being followed by the wet year of 2010 (Figs. 9b and 9c). Despite this, continuously-growing dryness expressed by cumulative scPDSI anomalies was well pronounced in 2007–2009, and it improved only slightly in the following year 2010. But the total situation was negatively influenced especially by the disastrous Kyrill windstorm on 18–19 January 2007, followed by Emma on 1–2 March 2008 (Hostýnek et al., 2008; Brázdil et al., 2018b), which were reflected in wind-related salvage felling of 8.842 million m³ of 400 wood in 2007 and 4.855 million m³ in 2008, i.e. 13.697 million m³ together.

### 4.2.7 The bark beetle outbreak of the 2010s–2020s

According to bark beetle salvage felling, the outbreak of 2015–2021 affected a broad belt of the districts spanning from the Šumava Mts. easterly to Moravia and turning there to the Jeseníky Mts. region, where in the Bruntál district felling reached 4.845 million m³ of wood (Fig. 10a). But the most affected area was the Bohemian-Moravian Highlands with districts 405 located there or in its broader neighbourhood (Jihlava 3.517 million m³, Jindřichův Hradec 2.746 million m³, Blansko 1.928 million m³, Havlíčkův Brod 1.887 million m³, Třebíč 1.883 million m³, Žďár nad Sázavou 1.824 million m³). Other highly affected districts consisted also of Písek in southern Bohemia (1.817 million m³) and Děčín in northern Bohemia (1.776 million m³). The bark beetle felling overcame 1 million m³ of wood in a further five districts in Bohemia and a further six districts in Moravia and Silesia. The total amount of bark beetle salvage felling in the whole CR during 2015–2021 reached 410 55.46 million m³ of wood (Fig. 2).

Temperature patterns during 2015–2021 were characterised by highly positive anomalies compared to the 1961–1990 reference (with especially warm summers), which was expressed in continuously growing cumulative monthly anomalies





(Figs. 10b and 10c). Annual precipitation totals were deeply below the mean in 2015 and 2018 when cumulative monthly anomalies reached *c.* –250 mm from autumn 2018 to spring 2020. Prevailing negative precipitation anomalies combined with increasing temperatures resulted in growing dryness expressed by cumulative scPDSI anomalies. Above-mean precipitation in 2020 and mean totals in 2021 did not improve dry patterns too much. The heavy windstorm Herwart on 29 October 2017 (Hujslová and Šimandl, 2018) complicated the situation by correspondingly high salvage felling in 2018 (4.62 million m³). But already from 2016 volumes of annual salvage felling due to strong winds were deeply below the related annual volumes for bark beetle felling (Fig. 10b).

### 4.2.8 Composite analysis of outstanding bark beetle outbreaks

In order to characterise general climatological conditions of all seven outstanding bark beetle outbreaks together, mean temperature, precipitation and scPDSI anomalies of five years (from n–5 until n–1) before the first year of an outbreak (n) and five years afterwards (from n+1 until n+5) were analysed. The years 1834, 1870, 1949, 1982, 1992, 2007, and 2015 were taken as the starting years of seven selected outbreaks. The analysis was performed separately for annual and summer series expressed as anomalies from the 1961–1990 reference period (Fig. 11). The highest and significant (p< 0.05) annual and summer temperatures in the starting year of the bark beetle outbreak are followed by especially warmer summers in the following years (significant in summers n+1 and n+3), but positive temperature anomalies compared to the reference period also appeared in the years n–1 to n–4 (Fig. 11a).

As for precipitation anomalies, they do not show such consistent patterns as temperatures do: the driest starting year of the outbreak was accompanied by those in n+1 and n–2 (significantly lower for both annual and summer values), while especially wet patterns occurred in the years n–5 and n+5 (Fig. 11b). Most consistent patterns appear in a composite of annual and summer scPDSI anomalies, indicating growing dryness from the year n–3 to its maximum in the n+1 year, and after slightly smaller scPDSI in the following year n+2, an important weakening of dryness afterwards, i.e. from n+3. Both summer and annual values show significant negative deviations for five years centred around the starting outbreak year, that is, from n–2 to n+2 (Fig. 11c). This clearly demonstrates the importance of temperature and precipitation patterns (and scPDSI as their combined effect) at the start of and two years following bark beetle outbreaks. But the triggering role of large windstorm calamities cannot be omitted here.

In order to analyse circulation patterns in the above key years of bark beetle outbreaks, we used an objective classification of circulation types (see Sect. 3) for the years n–1 to n+2 to compare composite relative frequencies of circulation types with their means in 1961–1990. Positive differences between them indicate higher relative frequencies of types in composite sets compared to those in the 1961–1990 reference period and *vice versa*. Because of the availability of objective classifications from 1961, we did composite analysis only for four selected outbreaks in the 1980s, 1990s, 2000s, and 2010s.

A composite of summers in the years n–1 to n+2 (Fig. 12) indicates statistically significant positive differences for the group of anticyclonic types (14.6 % in n+1 and 13.5 % in n+2), compensated by significant negative differences for directional types in the summers n to n+2 as well as cyclonic types for n+1 and n+2 summers. As for individual circulation types, the





type centered anticyclone A shows significant positive differences (maximum 5.4 % in n+2) as well as other anticyclonic types with westerly airflow (maxima: anticyclonic southwestern ASW 4.9 % in n+1, anticyclonic western AW 4.9 % in n+2, anticyclonic northwestern ANW 3.9 % in n+1) (for schemes of these types see Fig. S2). These types are favourable for warm and dry weather in summer. For statistically significant differences in individual cyclonic circulation types there are

characteristic rather negative deviations, achieving values primarily between –1 % and –2 %, despite the fact that during the first summer of bark beetle outbreaks the types CE (cyclonic eastern), CSE (cyclonic southeastern), and CNW (cyclonic northwestern) experienced positive differences. Among directional circulation types with significant differences, those with colder airflow from the north or east directions like the eastern type E –2.6 % in n+2 summer (as significant also in n–1 and n+1) or the northern type N with –4.0 % in the summer n and –2.6 % in n+2 especially appear. A similar dominance of

anticyclonic types A, ASW, AW, and ANW, as in the summer, appears also in the annual composite, when cyclonic and directional types show again rather negative differences; but the number of circulation types with statistically significant differences is considerably smaller (not shown).

## 5 Discussion

### 5.1 The database and data uncertainty

Data concerning past and recent bark beetle outbreaks are biased by different types of uncertainties. As is typical for data collected from different documentary sources (e.g., Brázdil et al., 2005), they can only go into certain detail (e.g., forest stands belonging to any estate in the past without detailed specification into individual forest districts) and suffer by incompleteness, i.e. they represent rather lower estimates of spatiotemporal occurrences. Many reports of past bark beetle outbreaks suffer from missing quantitative data about the volume of damaged wood. Such data appears rather randomly in

some reports without the possibility to obtain total volumes for the whole country. If such country estimates do exist, they can differ from author to author, or only repeat earlier presented data. Concerning older reports, in some cases not only could *Ips typographus* have been named as a primary pest damaging spruce trees, but information could also include other pests or conifers (fir, pine), depending also on the related development of knowledge about forest pests (see Sect. 5.2). For example, the order of Prague *Gubernium* from 1784 "points out a mass withering of spruce and fir caused by worms between wood

and bark, which cannot be killed …" (Nechleba, 1929), also reported them as "flying worms" (Lubojacký, 2012).

Concerning data about bark beetle salvage felling from 1964, some uncertainties may also appear there. It concerns, for example, problems with a clear distinction between the volumes of salvage felling attributed to drought, air pollution, and bark beetle infestation, as reported by individual foresters. Concerning the homogeneity of the whole 1964–2021 series, political change in former Czechoslovakia after the Velvet Revolution in 1989 led to a change in the ownership of forest

stands. The privatization of some forests (from former state ownership to private owners) after 1990 led to a lower number of reporting districts, i.e. data concerns approximately three-quarters of the forest stands in the CR.



Despite the above uncertainties, the created database, covering the last 240 years, represents a unique source of information not only for the CR, but also as an important contribution to the knowledge of past (historical) and present bark beetle outbreaks in Europe, complementing similar data from some other studies (see e.g. Fig. 8 in Schelhaas et al., 2003). For

example, Schafstall et al. (2022) found only three main bark beetle outbreaks between 1400 BP and the present in the High Tatras (Slovakia) (the last one post-2004 after the disastrous windstorm Elizabeth of 19 November 2004), in which *Ips typographus* was rather low in numbers. Moreover, bark beetles remain rather neglected in palaeoecology with only a few dozen reported sites in Europe for the whole of the Holocene (Schafstall et al., 2020).

**5.2 Responses to past bark beetle outbreaks**

Studies on bark beetles were published as early as the 18th century, especially by foresters from Saxony, which was hit by a severe bark beetle outbreak in the 1780s (Jäger, 1798; Dallinger, 1798), similar to the Czech Lands (see Sect. 4.1.1). These works also mention that beetles had been destroying spruce stands for centuries, for example, in the Harz mountains in north-central Germany. Similarly, in the Czech Lands, Trzebitzky (1798) published a practical handbook on the drying out of needle-leaved forests, in which he discussed bark beetles as well. Bark beetles also remained a much-discussed topic in

the first half of the 19th century (e.g., Krutzsch, 1825), but, according to Pfeifer (1875), in the 1820s there was still a general lack of deeper knowledge among local foresters about bark beetles. Such early studies mainly focused on the natural history of bark beetles in order to come up with practical advice for forestry managers in preventing outbreaks. The 1833 outbreak led to the publication of management guidelines by the Czech *Gubernium* on the handling of beetle infected forest stands (Nožička, 1957).

The massive outbreak following the 1868 and 1870 windstorms led to intense interest in the topic. First, the logistics of removing huge amounts of infected trunks posed considerable challenges. For example, in the Šumava Mts., the processing of damaged trees required the construction of new roads and tracks. To obtain access to the forests, 33,000 trunks in the Vimperk area had to be processed in November 1870 (Záloha, 1970). Moreover, seven water sawmills were newly created and 850 workers from Bohemia and Bavaria were hired, of whom 670 worked throughout the summer of 1871. However, it

was clear that infected trees could not be removed quickly enough. This created societal tensions as well. An attempt to save money for the hiring of foreign workers after the 1870 windstorm was reported on the Prášily estate. The local forester hired only 50 foreign workers to process the windstorm calamity and, together with local workers, they were unable to debark all trunks in time and a subsequent bark beetle outbreak threatened the forests of surrounding estates, including those in neighbouring Bavaria (US8, pp. 217–218). The owner of the Chodová Planá estate warned his forest employees in 1875 that

they were not allowed to speak publicly about the calamity and that all the data about the calamity felling was an official secret. This was a reaction to newspaper articles criticising the estate for its insufficient measures against bark beetles (US9, pp. 117–118).

The catastrophe also shook the foundations of the forestry profession. The Society of Bohemian Foresters (*Böhmischer Forstverein*) – the main forum for exchanging knowledge on forestry in the country – kept the topic on its agenda. Lively

discussions on bark beetles took place at the plenary meetings of the Society every year from 1870 to 1875. In the beginning,
      most contributions focused on technical solutions, such as the usage of 'trap trees' (to attract bark beetles) or the swift
      removal and burning of infected tree bark. Only occasional references were made to alternative methods; for example, the
      protection of birds, being as they are the natural enemies of bark beetles (Vereinsschrift, 1872, p. 64). In 1874, however,
      *Forstmeister* Alois Nedobitý argued that the outbreak could to a large extent be blamed on the foresters themselves. The

forced removal of dead trees was counterproductive because it prevented those useful insects from developing that could
      slow down the procreation of bark beetles (Vereinsschrift, 1874, p. 66). A year later, *Forstmeister* Soucha made an appeal to
      the entire nation (and, by implication, to all Czech foresters) to reconsider clear-cuts and tree planting as their preferred way
      of forest regeneration. What the bark beetle outbreak taught foresters, he argued, was that natural regeneration was plentiful
      under the damaged trees, if only foresters did not destroy it through tree removal (Vereinsschrift, 1875, pp. 150–157).

However, reactions to such contributions were not favourable and cutting remained the main method to combat bark beetle
      outbreaks.

      In the following decades, the focus of foresters shifted from bark beetles to other insects, mainly the nun moth, which caused
      much more damage in this period (Syrovátka, 1922). However, the catastrophes of the latter were used as opportunities to
      carry out comparative studies on bark beetles in the 1920s (e.g., Komárek, 1925). Some attention reverted to bark beetles as

a result of the outbreak in 1944–1947. New perspectives were also included: for example, Pfeffer (1952) wrote that this
      outbreak was the first that was not directly connected to the previous windstorm. Rather, the proximate cause was the
      excessive harvesting of spruces during World War II that were often left lying in forests.

      As the state gradually took over the management of forests from estates, regulations concerning bark beetles started to be
      included in pertinent national legislation. The first and second forest laws of Czechoslovakia (1960 and 1977) both included

passages on bark beetles, mainly on the obligations of forest owners to find and remove infected trees. Following the
      outbreak of the 1980s, special instructions for fighting bark beetles were issued by the Ministry for Forestry and Water
      Management (Lubojacký, 2012). *Mutatis mutandis*, all these instructions followed the general principles established in the
      19th century without questioning their general validity or effectiveness.

**5.3 Environment, climate, and bark beetle outbreaks**

Bark beetle outbreaks in the 19th century appear to have happened mostly at higher altitudes, where the presence of
      extensive spruce stands can be expected. Even though conifer plantations started in the Czech Lands after 1800 (Nožička,
      1968), discussions after the bark beetle outbreak in the 1870s suggest that many, if not most, stands affected were not yet
      spruce monocultures. More detailed studies would be needed to confirm this, but the prevailing method of dealing with
      calamities (removing all infected trees) as well as the artificial forest regeneration that – as observed by foresters already in

the 19th century – was the consequence of this method, may have contributed to creating forests that were increasingly
      susceptible to windstorm (Brázdil et al., 2018b) and also to bark beetle outbreaks. In addition, especially in the 20th century,
      spruce was extensively planted in even-aged monocultures in lower-lying regions as well. While such regions were not hit



by bark beetle in the 1940s–1950s (Fig. 6), they suffered considerable damage from the 1980s onwards (Figs. 7–10). The planting of extensive conifers monocultures reduced the biodiversity of Czech forests, exhausted the soil and increased the

susceptibility of forests to natural disasters (Daniel et al., 2013). However, in the last few years, the area covered by spruce monoculture plantations in the CR declined from 54.1 % in 2000 to about 48.8 % in 2020 (Ministerstvo zemědělství, 2021).

The analysis of past and recent large bark beetle outbreaks in the CR demonstrates the importance of meteorological (windstorms) and climatological (temperature and precipitation, resulting in droughts) factors as the main triggers of such outbreaks (cf. Fig. 11), appearing as a manifestation of long-term climate variability, especially enhanced by recent climate

change (cf. Fig. 3). These results are confirmed by many other European studies, demonstrating the importance of temperatures for the development and voltinism of bark beetles (e.g., Annila, 1969; Wermelinger and Seifert, 1998, 1999; Jönsson et al., 2009), heavy windstorms as drivers of subsequent bark beetle outbreaks (e.g., Temperli et al., 2013; Thom et al., 2013; Stadelmann et al., 2014; Seidl and Rammer, 2017; Brázdil et al., 2018a) and droughts accompanied by high temperatures as other key factors in producing disastrous outbreaks (e.g., Marini et al., 2017; Matthews et al., 2018; Netherer

et al., 2019; Hlásny et al., 2021b; Jaime et al., 2022).

Unsurprisingly, many researchers address the problem of bark beetle outbreaks in the context of future climate change manifested in global warming, the latter of which will contribute to the worsening of living conditions for spruce forests especially (e.g., Hlásny et al., 2011; Jönsson et al., 2011; Berec et al, 2013; Jakoby et al., 2019; Sommerfeld et al., 2020). Hlásny et al. (2021a) summarised the effects of climate change on bark beetles as being factors facilitating their survival and

development, increasing their potential to spread into higher altitudes and latitudes, and contributing to reduced tree resistance due to extreme weather/climatic extremes.

Recent climate development in the CR is characterised by continuous warming in past decades, enhanced from the 1980s and especially from the 2010s (Zahradníček et al., 2021, 2022), and relatively stable precipitation totals (Brázdil et al., 2021) with increasing soil dryness (Trnka et al., 2015; Řehoř et al., 2021b), and this creates suitable conditions for increasing

climatic stress on forest stands, resulting in disastrous bark beetle outbreaks. All described negative climatic consequences for Norway spruce plantation in the CR will be further enhanced by the fact that projections to 2041–2060 under the continuance of recent climate change indicate that suitable conditions for spruce only comprise 11.3 % of the area of Czech forests, while in 1961–1990 it was 46.0 % (Čermák et al., 2021).

The recent bark beetle devastation of forests in the CR destroyed in seven years from 2015 to 2021 55.46 mil. $m^3$ of wood

altogether, which is more than the total for the previous 51 years (34.80 mil. $m^3$) for which quantitative bark beetle salvage felling data are available (cf. Fig. 2). The highest annual bark beetle felling in 2020 (14.89 million $m^3$) was not exceeded by any other biotic factors (other insects, gnawing of trees by wild animals, mushroom pathogens) nor by any abiotic factors (wind, snow, rime, drought, air pollution). After three previous bark beetle outbreaks, it was also the first time when bark beetle salvage felling was higher than those caused by windstorms (17.03 million $m^3$ in 2015–2021).

Besides environmental, cultural, and socio-economic consequences, the recent devastation of spruce forest stands and the subsequent formation of forest hollows due to the felling of trees influence fluxes of $CO_2$ between forests and the



atmosphere, when places cleared of trees enhance fluxes of $CO_2$ from the soil to the atmosphere (Hlásny et al., 2021a). This means that such stands can change from being $CO_2$ sinks to $CO_2$ sources, as has already occurred from 2018 in the CR (e.g., CENIA, 2021), and in such ways indirectly impacting on the climate by producing this important greenhouse gas (13.6 Mt

$CO_2$ eq. for 2019 in LULUCF – Land Use, Land-Use Change and Forestry). As follows from inventorying greenhouse gases produced in the CR, the forests have become sources of increasing $CO_2$, further promoting the "anthropogenic" footprint of GHG emissions due to the deterioration of the health conditions of forests caused by bark beetle infestation and droughts (e.g., CENIA, 2021).

## 6 Conclusion

From a systematic analysis of the past and present bark beetle outbreaks in the CR for the 18th–21st centuries, the following conclusions can be summarised:

(i) Different types of documentary sources and bark beetle salvage felling were used to create a unique database of the past bark beetle occurrences and their impacts on the territory of the CR from the 18th century to the present. The database allowed us to analyse the spatiotemporal variability of bark beetle outbreaks and select the most outstanding bark beetle

outbreaks from 1781 CE.

(ii) Disastrous bark beetle outbreaks of the 19th century in the 1830s and 1870s were triggered by extreme windstorms, when it was not possible to quickly remove a great amount of blowdown wood from forest stands, thus creating good conditions for bark beetle expansions. The drought episodes following windstorm disturbances intensified devastating bark beetle effects.

(iii) Disastrous bark beetle outbreaks in the 1980s, 1990s, 2000s, and 2010s are clearly attributed to recent climate change triggered by the compound effect of increasing temperatures and relatively stable precipitation totals, resulting in extreme droughts when many spruce stands now appear to be located in unsuitable environmental conditions, being more susceptible to different disturbances such as those by the bark beetle.

(iv) The last bark beetle outbreak in the CR from the mid-2010s forced by high temperatures and drought has no documented

historical analogue to its scope and its devastating effects on Czech spruce forests. It has contributed to the fact that Czech forests have become from 2018 sources of $CO_2$ increase, further increasing the "anthropogenic" footprint of GHG emissions.

(v) Despite great attention devoted to the recent and potential future bark beetle outbreaks forced by recent climate change, there is a great potential to learn more from the past. This study demonstrates a high potential of documentary evidence for the creation of similar databases for other European regions to better understand the natural and socio-economic

development that led to the recent situation.

**Data availability.** The datasets and series used in this article are not publicly available and can be obtained only on the personal request.



**Author contributions.** RB extended bark beetle chronology, designed and wrote the paper with contributions from all
coauthors. PZ provided all data related to bark beetle and abiotic salvage felling a prepared the first version of bark beetle
chronology. PS dealt with responses to bark beetle outbreaks. KC worked with analysis of salvage felling together with
climatic conditions and finalised all figures. PD contributed with composite analysis. LD collected digital copies related to
bark beetle information from Forest History Research. MT calculated scPDSI series and contributed to relationship of bark
beetle forest devastation and $CO_2$ production. JŘ analysed relationship of more recent bark beetle outbreaks to circulation
types. SS excerpted bark beetle data from Neue Schriften. All authors have read and commented on the latest version of the
paper.

**Competing interests.** The authors declare that they have no conflict of interest.

**Special issue statement.** This article is not a part of the special issue. It is not associated with a conference.

**Acknowledgements.** RB, PD, LD and MT were supported by the Ministry of Education, Youth and Sports of the Czech
Republic for the SustES – Adaptation strategies for sustainable ecosystem services and food security under adverse
environmental conditions (project no. CZ.02.1.01/0.0/0.0/16_019/0000797), PZ by the Ministry of Agriculture of the Czech
Republic, institutional support MZE-RO0118, KC by Global Change Research Institute of the Czech Science Academy and
JŘ by Masaryk University within the MUNI/A/1393/2021. We acknowledge Petr Štěpánek (Brno) for Czech temperature
and precipitation series and Laughton Chandler (Charleston, SC) for English style corrections.

**Financial support.** This research has been funded by the Ministry of Education, Youth and Sports of the Czech Republic
(grant no. CZ.02.1.01/0.0/0.0/16_019/0000797).

**Unpublished sources**

Ústav pro hospodářskou úpravu lesů, Brandýs nad Labem: Historický průzkum lesů (Forest History Research).

(US1) Málek, J. (1965): Lesní závod Telč (Forest plant Telč).

(US2) Tlapák, J. (1960): Lesní závod Červené Poříčí (Forest plant Červené Poříčí).

(US3) Ministr, J. (1964): Lesní hospodářský celek Bečov I a II (Forest economic complex Bečov I and II).

(US4) Ministr, J. (1963): Jednotný hospodářský celek Kašperské Hory I a II (United economic complex Kašperské Hory I
and II).

(US5) Ministr, J. (1960): Jednotný hospodářský celek Přísečnice (United economic complex Přísečnice).

(US6) Horák, K. (1968): Lesní hospodářský celek Choceň II. Lesní závod Vysoké Chvojno (Forest economic complex
Choceň II. Forest plant Vysoké Chvojno).



(US7) Hošek, E. (1961): Lesní hospodářský celek Maršov a Trutnov (Forest economic complex Maršov and Trutnov).

(US8) Stolařík, R. (1974): Lesní hospodářský celek Modrava (Forest economic complex Modrava).

(US9) Ministr, J. (1969): Lesní hospodářský celek Planá I. II. III. Lesní závod Planá u Mar. Lázní (Forest economic complex Planá I, II, III. Forest plant Planá at Mariánské Lázně).

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





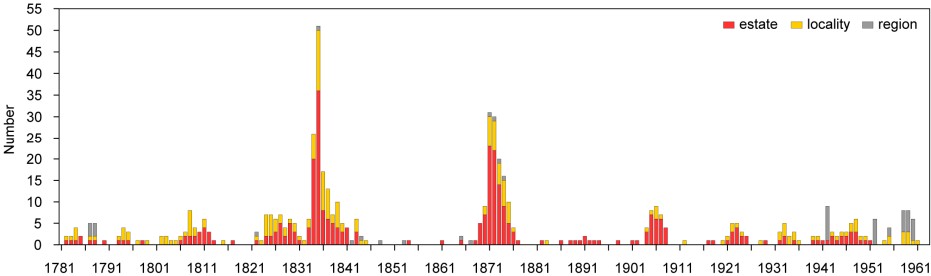

**Figure 1: The annual numbers of locations with bark beetle outbreaks over the territory of the Czech Republic during the 1781–1963 period extracted from documentary evidence.**






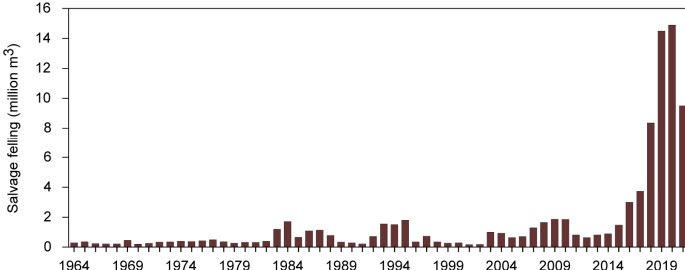

**Figure 2: Fluctuations in bark beetle salvage felling (million m³ of wood) for the Czech Republic in the 1964–2021 period (source of data: Research Institute of Forestry Economics and Gamekeeping).**





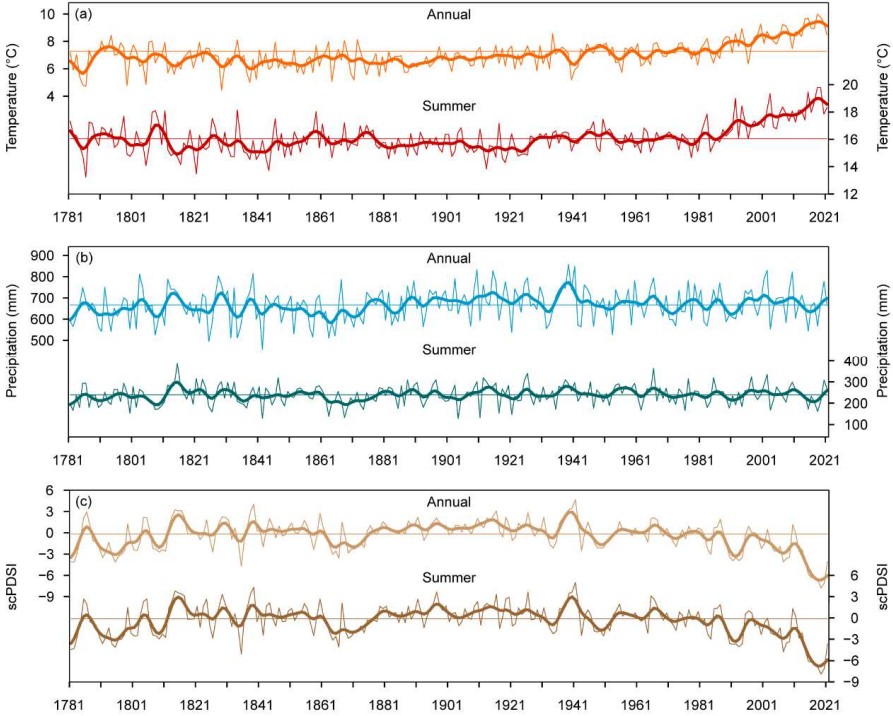

**Figure 3: Fluctuations in annual and summer series of (a) mean air temperatures, (b) precipitation totals, and (c) scPDSI for the territory of the Czech Republic during the 1781–2021 period. The values are smoothed by a 10-year Gaussian filter (bold line) and complemented by horizontal lines corresponding to related means of the 1961–1990 period.**



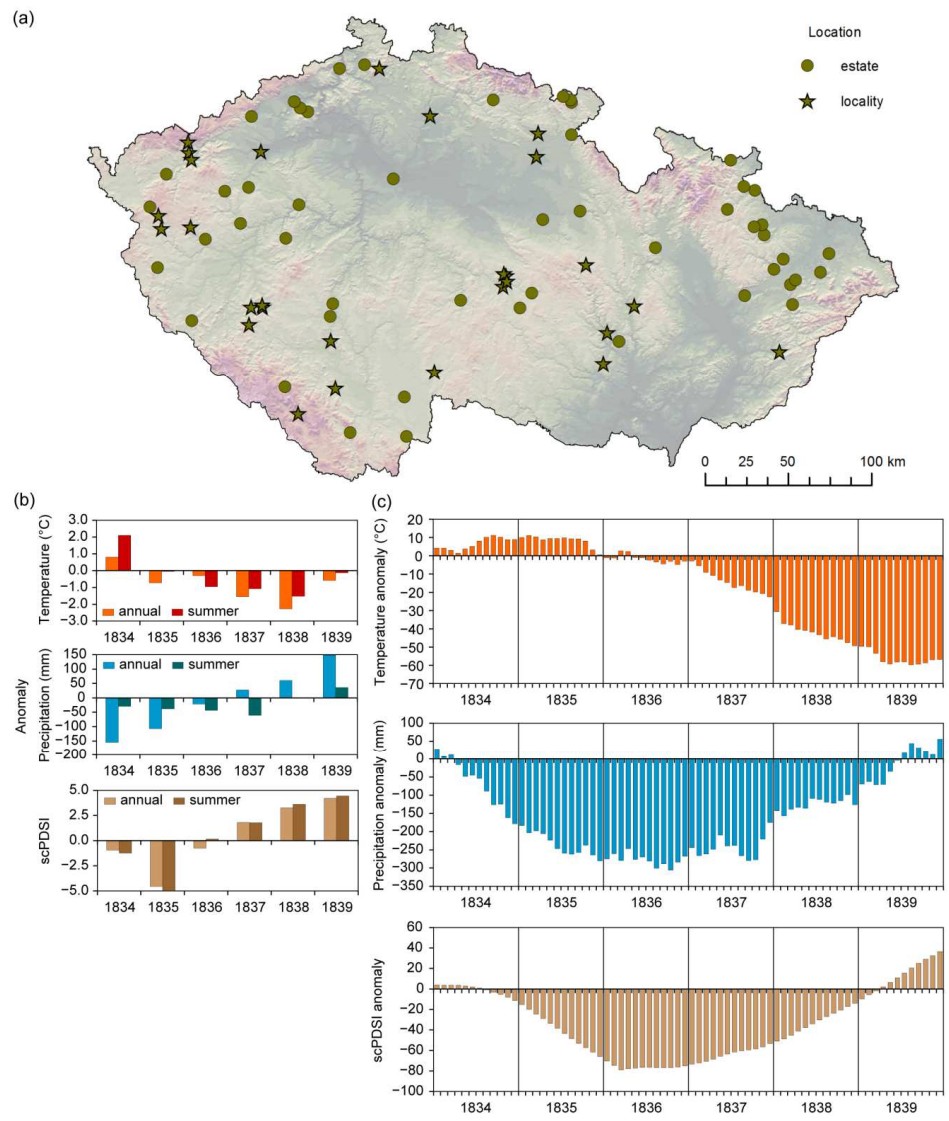

**Figure 4: Bark beetle outbreak of the 1830s on the territory of the Czech Republic: (a) locations with the occurrence of bark beetle; (b) annual and summer anomalies of temperature, precipitation, and scPDSI in 1834–1839; (c) cumulative monthly anomalies of temperature, precipitation, and scPDSI in 1834–1839 (1961–1990 reference period).**


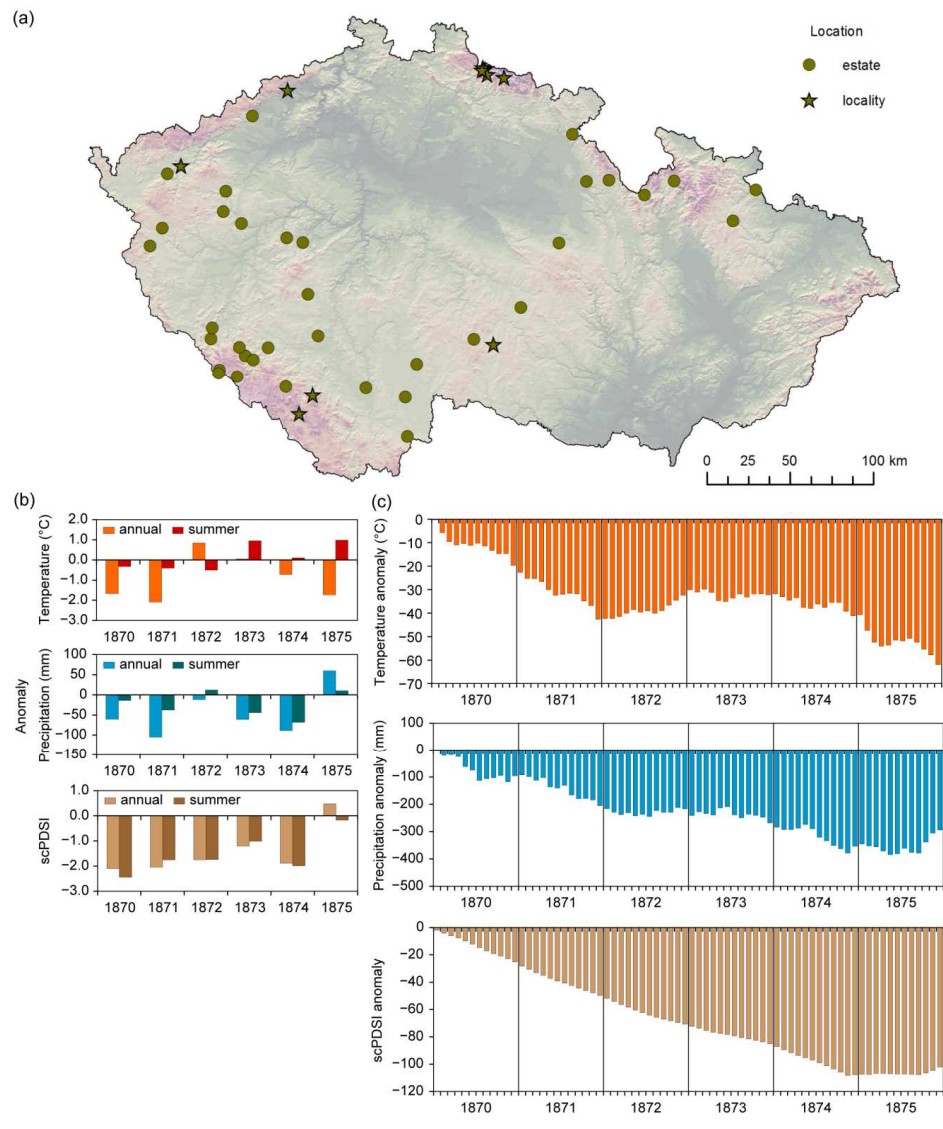

**Figure 5: Bark beetle outbreak of the 1870s on the territory of the Czech Republic: (a) locations with the occurrence of bark beetle; (b) annual and summer anomalies of temperature, precipitation and scPDSI in 1870–1875; (c) cumulative monthly anomalies of temperature, precipitation and scPDSI in 1870–1875 (1961–1990 reference period).**



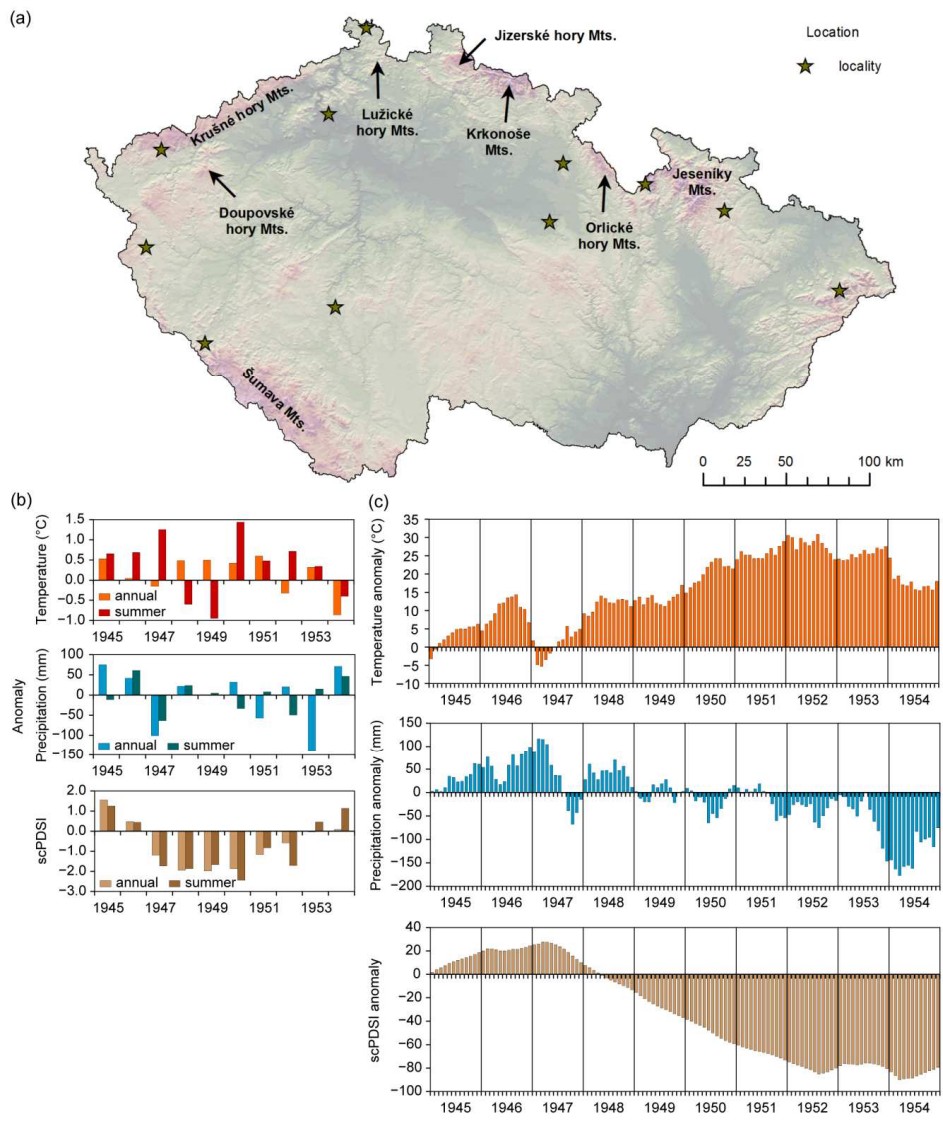

**Figure 6: Bark beetle outbreak of the 1940s–1950s on the territory of the Czech Republic: (a) location of regions and localities with the occurrence of bark beetle; (b) annual and summer anomalies of temperature, precipitation and scPDSI in 1945–1954; (c) cumulative monthly anomalies of temperature, precipitation and scPDSI in 1945–1954 (1961–1990 reference period).**



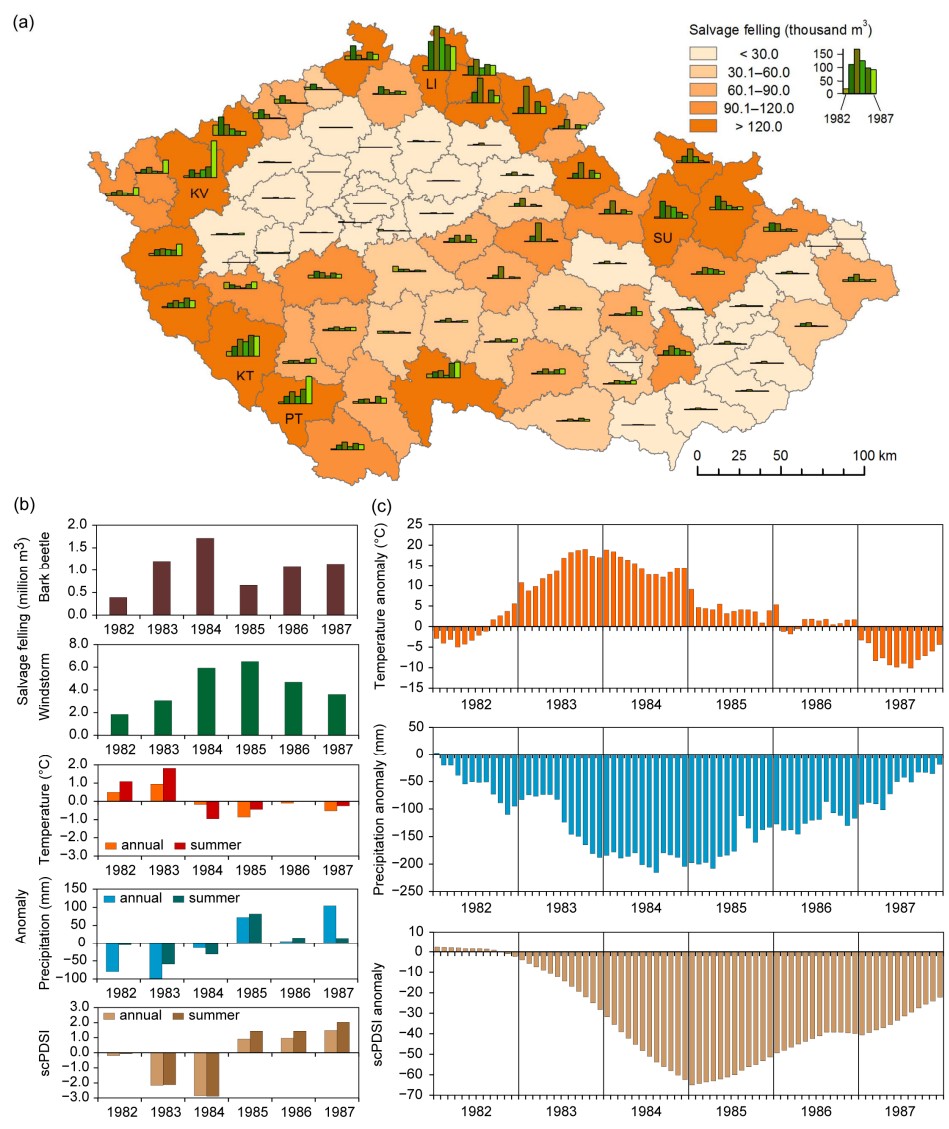


**Figure 7: Bark beetle outbreak of the 1980s on the territory of the Czech Republic: (a) the volume of bark beetle salvage felling in 1982–1987 in individual districts; (b) annual bark beetle and windstorm salvage felling, annual and summer anomalies of temperature, precipitation and scPDSI in 1982–1987; (c) cumulative monthly anomalies of temperature, precipitation and scPDSI in 1982–1987 (1961–1990 reference period). Abbreviations of districts: LI – Liberec, KT – Klatovy, KV – Karlovy Vary, PT –**
**Prachatice, SU – Šumperk.**

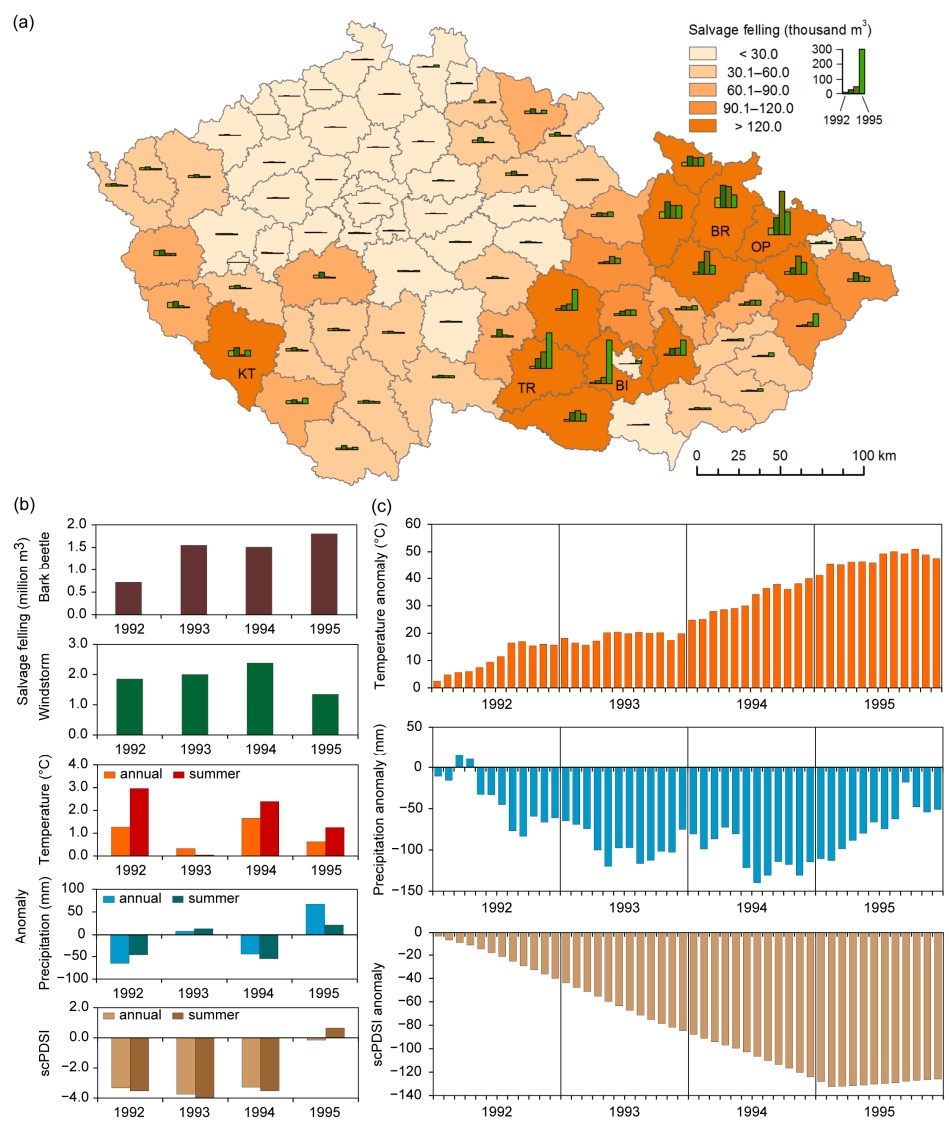

**Figure 8: Bark beetle outbreak of the 1990s on the territory of the Czech Republic: (a) the volume of bark beetle salvage felling in 1992–1995 in individual districts; (b) annual bark beetle and windstorm salvage felling, annual and summer anomalies of temperature, precipitation and scPDSI in 1992–1995; (c) cumulative monthly anomalies of temperature, precipitation and scPDSI in 1992–1995 (1961–1990 reference period). Abbreviations of districts: BI – Brno-venkov, BR – Bruntál, OP – Opava, KT – Klatovy, TR – Třebíč.**



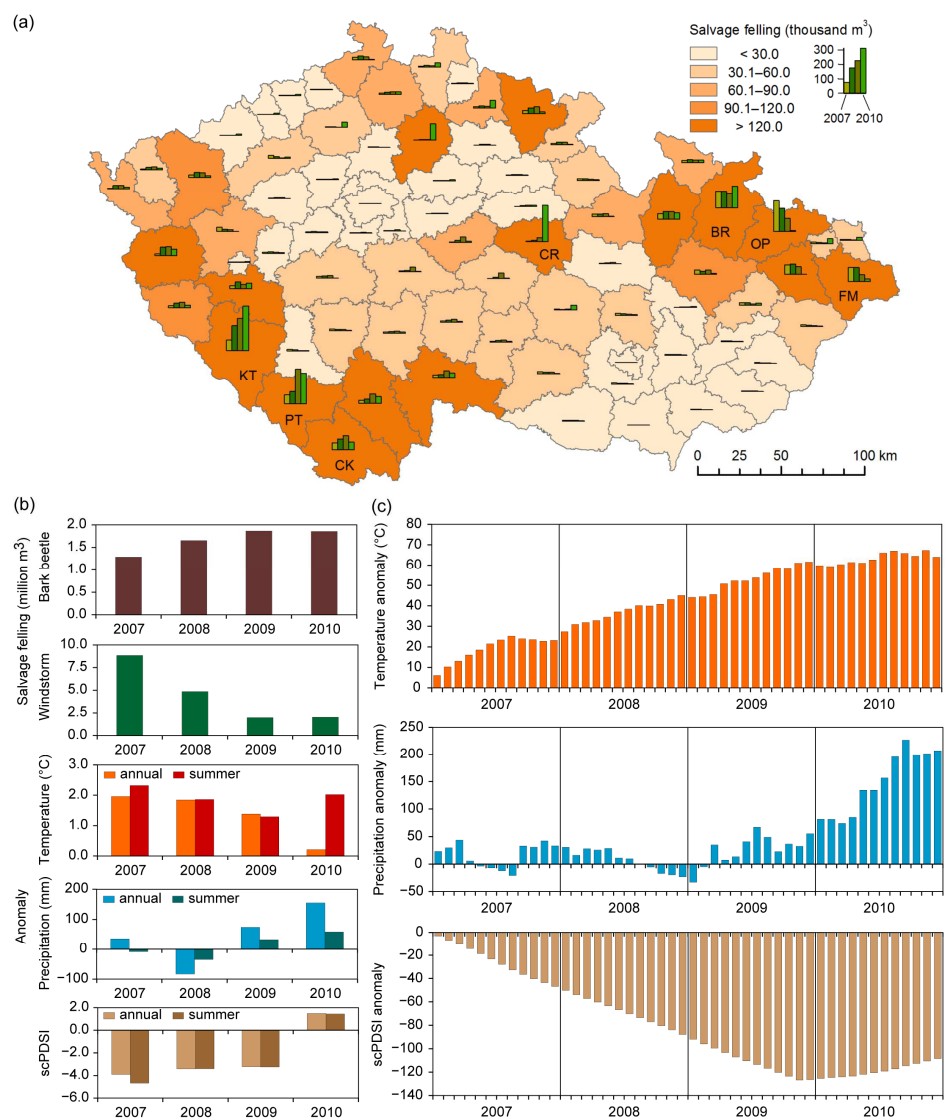

**Figure 9: Bark beetle outbreak of the 2000s on the territory of the Czech Republic: (a) the volume of bark beetle salvage felling in 2007–2010 in individual districts; (b) annual bark beetle and windstorm salvage felling, annual and summer anomalies of temperature, precipitation and scPDSI in 2007–2010; (c) cumulative monthly anomalies of temperature, precipitation and scPDSI in 2007–2010 (1961–1990 reference period). Abbreviations of districts: BR – Bruntál, CK – Český Krrumlov, CR – Chrudim, FM – Frýdek-Místek, OP – Opava, KT – Klatovy, PT – Prachatice.**


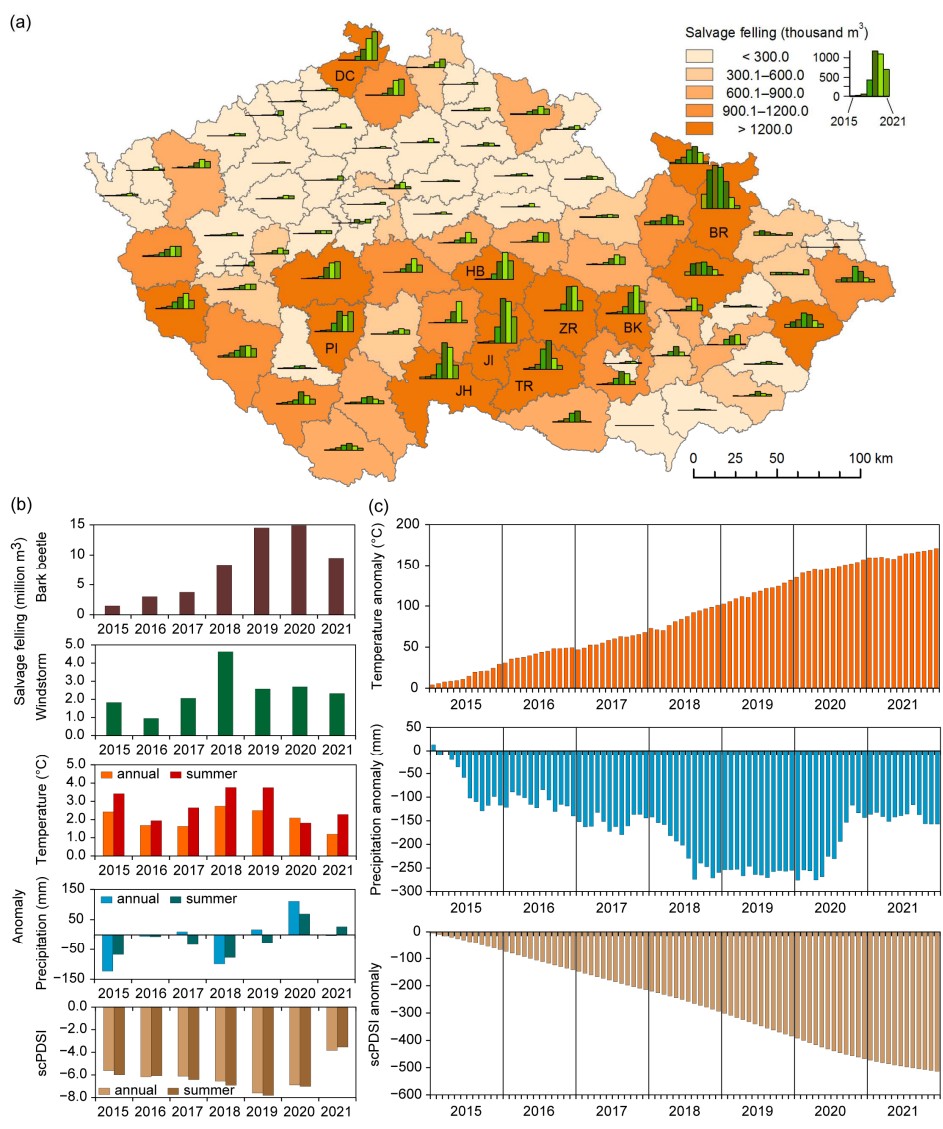

**Figure 10:** Bark beetle outbreak of the 2010s–2020s on the territory of the Czech Republic: (a) the volume of bark beetle salvage felling in 2015–2021 in individual districts; (b) annual bark beetle and windstorm salvage felling, annual and summer anomalies of temperature, precipitation and scPDSI in 2015–2021; (c) monthly cumulative anomalies of temperature, precipitation and scPDSI in 2015–2021 (1961–1990 reference period). Abbreviations of districts: BK – Blansko, BR – Bruntál, DC – Děčín, HB – Havlíčkův Brod, JH – Jindřichův Hradec, JI – Jihlava, PI – Písek, TR – Třebíč, ZR – Žďár nad Sázavou.





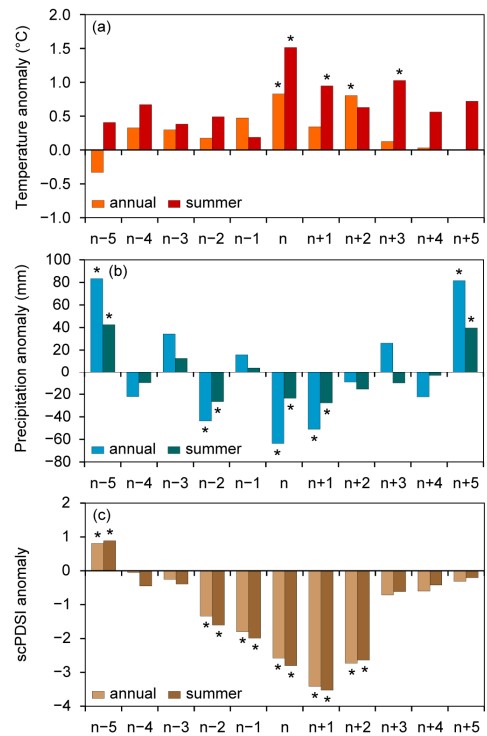

**Figure 11: Composite of annual and summer temperatures (a), precipitation (b) and scPDSI (c) in five years before (from n–5 until n–1) and five years after (from n+1 until n+5) the starting year (n) of bark beetle outbreak for the seven outstanding outbreaks of the 19th–21st centuries on the territory of the Czech Republic. Values are anomalies with respect to 1961–1990 and significant anomalies (p<0.05) are indicated with asterisks.**



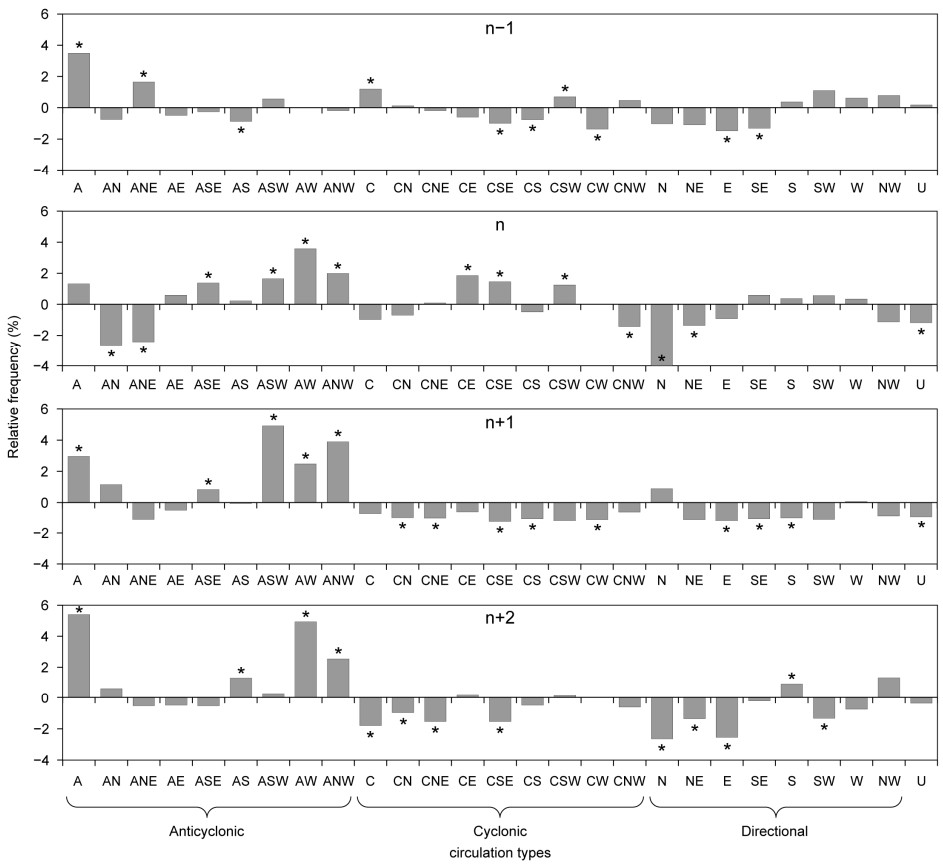

**Figure 12: Differences in mean relative frequencies of individual circulation types of the objective classification in a composite of summers related to four bark beetle outbreaks from the 1980s, and mean relative frequencies of the 1961–1990 reference period in the Czech Republic: n–1 – a year before the first year of a bark beetle outbreak (n) and two years n+1 and n+2 after the first year n. Symbol * indicates statistically significant differences.**