# Peer review of "Meteorological and climatological triggers of past and present bark beetle outbreaks in the Czech Republic"

_Climate of the Past, 2022_

## Referee Comment (RC2)

[referee-annotated manuscript omitted]

---

## Author Comment (AC1)

**Responses to the referee 1**

The manuscript reviews bark beetle (BB) outbreaks in Czech Republic for the last 240 years. The authors collated much data of different quality on such BB gradations and related the damage wood to meteorological extreme events. While such gradations usually followed windstorms in the 19th century, the recent and most devastating gradation was primarily related to warmer and drier weather prevailing during the last decade in the frame of climate change. Authors highlight the importance documentary databases with rich information on development of the environment and the socio-economy that foster the understanding of current forest damages.

I found the paper too long, the descriptions too detailed (though very interesting) and the conclusions too poor to be published as a concise paper. There are several ways to cope with this manuscript: to allow a long paper like this, or to improve the manuscript by reducing the information to digestible texts and figures, while putting details into the supplementary material. I propose the second treatment, by focusing on the following aspects:

RESPONSE: We would like to thank the referee 1 for evaluation of our manuscript and many critical comments we are trying to respond below.

Language: Much of the content can be improved an English brush-up and by more precise writing. I had much difficulties in the first half or the text and less so in the rest. Perhaps, the first part was not treated with the same care than the second part. Just my impression.

RESPONSE: The paper was corrected by a native speaker, Laughton Chandler (Charleston, SC). Further changes concerning English were included based on the annotated manuscript by the second native referee, Neil Macdonald. We believe that after these corrections, the English style of our paper is already acceptable. Otherwise, we would very appreciate the marking of particular shortcomings in the manuscript.

Descriptive event history on BB gradations: The information could be put in a chronologically ordered table, listing the most important triggers and forestry concepts of fighting against BB presence and also mentioning windthrow events.

RESPONSE: We developed a new database of bark beetle outbreaks which consists of 11 pages of very detailed information (without data sources, which represent 5 additional pages). This database information is reduced in Fig. 1 into frequency of places with bark beetle outbreaks and in Fig. 2 with bark beetle salvage felling to select the most notable past and present bark beetle outbreaks. Because we plan to carry out further analyses based on this database, we would not like to make it publicly available (only upon request with specification of corresponding conditions of its use). Concerning windthrow events, we added to Figure 3 a series of severe windstorms across the Czech Republic derived from documentary and instrumental data. Concrete cases of severe windstorms that contributed to notable bark beetle outbreaks are mentioned in each outbreaks analysed. A new Figure 13 comparing bark beetle outbreaks with windstorms was added with a corresponding description to Section 5.3.

Figures 7-10: Could be put into a supplementary material. Instead, one figure would do to show the length of the BB gradation (in lines) and the impact of windstorms on salvage felling. Here, the big impact of the 2018 storm Burglind could be highlighted, which presumably strongly amplified the BB gradation.

RESPONSE: This paper has a regional character. It supposes that it should contain particularly detail regional evidence. If we are analysing particularly seven selected "notable" bark beetle outbreaks, what we see as important results of the article, moving Figures 7-10

with the most detail and highest quality data into supplementary material would mean real degradation of this paper. From this reason we would like to preserve these figures in the main text. Concerning of the 2018 storm Burglind, it affected particularly Ireland, the U.K., France, Netherlands, Belgium, Luxembourg, Germany, and Switzerland, but not the territory of the Czech Republic.

Windstorms: It is unclear to me why the authors claim that windstorms were only important in the 19th century. Rather, by looking at figures 7–10, it appears that such storms (usually winter storms) had quite an important role as a trigger. But the information is simply missing, i.e. the exact date of the strongest winterstorms (compare e.g. Usbeck et al. 2010, Agric. For. Meteorol. 150).
RESPONSE: While two notable bark beetle outbreaks of the 19th century followed after disastrous windstorms (i.e., they were their primary triggers), events in Figures 7-10 had as the main meteorological trigger warm and dry patterns, with clear coexistence of severe windstorms in 1984 (Figure 7) and 2007-2008 (Figure 9), as mentioned in the corresponding sections. The role of windstorms was pointed out also in two other cases. Dates of strongest windstorms are reported there. We corrected also one of conclusions describing the compound effect of several meteorological factors – see next point.

Coniferous vs. deciduous tree species: I could not find information on the percentage of forests in CR that represent natural forest communities. I suspect that a high proportion of the coniferous forests are planted since perhaps 200 years. BB infestations could be seen as an interaction of non-natural forest trees, windthrow (mainly in wintertime when coniferous trees are more susceptible than deciduous trees) and warmer/dryer weather in the frame of climate change. The conclusion that "many spruce stands now appear to be located in unsuitable environmental conditions" could be highlighted and substantiated with more detail.
RESPONSE: The corresponding point of conclusions was changed as follows: "(iii) Disastrous bark beetle outbreaks in the 1980s, 1990s, 2000s, and 2010s are attributed to recent climate change triggered by the compound effect of windstorms, strongly increasing temperatures and relatively stable precipitation totals of the past decades, resulting in extreme droughts, when many spruce stands now appear to be located in unsuitable environmental conditions, being more susceptible to different disturbances such as those by the bark beetle."
It is difficult to amount the precise percentage of natural forest communities because most of the territory of the Czech Republic was affected by human activities in the past. According to the Report on the State of Forests and Forestry in the Czech Republic in 2020, natural forests represent only 373,000 ha (4.7%) of the Czech territory. As we stated in lines 536–537, "conifer plantations started in the Czech Lands after 1800 (Nožička, 1968)". Such data support your assumption about bark beetle infestations as interaction with non-natural forest trees. However, several bark beetle outbreaks in the last decades took place in natural parks (e.g. in the Šumava Mts.) representing 1.8% of all natural forests. Bark beetle outbreaks do not avoid even natural forests.

L28: «negative impacts» Impacts on what? On vegetation? On forests? On Nature?
RESPONSE: Deleted. Please see the new version of the related sentence.

L29: overpopulation: gradation?
RESPONSE: Deleted. Please see the new version of the related sentence.

L30: More specific here?  ...caused a catastrophic Norway spruce decline with xxx ha of spruce stands affected.

RESPONSE: Information of the infested area is not available (we are working with volumes of salvage felling caused by bark beetle).

L31: Again, not very precise: Huge bark-beetle infestations followed extraordinary hot-dry episodes in summertime that reduced the vitality of Norway spruce...
RESPONSE: Accepted, the corresponding part of the text was changed as follows: "The reason is that dry episodes with high temperatures decrease the vitality and structure of spruce which has shallow roots, these characteristics result in less resistance to bark beetle attacks. In addition, droughts with high temperatures accelerate bark beetle evolution and increase the number of generations per breeding season (summer) to three in lower altitudes and two in higher ones (e.g., Pfeffer and Skuhravý, 1995; Jönsson et al., 2009, 2011; Netherer et al., 2019; Hlásny et al., 2021a)."

L32: In addition, (not „on the other hand")
RESPONSE: Accepted and corrected.

L40-65: This can be reduced to few lines with the same information remaining.
RESPONSE: Accepted, we changed a corresponding part of the text as follows: "Among different aspects of bark beetle outbreaks, several papers analysed in great detail the effect of temperatures on the duration of the egg, larval, and pupal stages of the spruce bark beetle and adult maturation feeding (e.g., Annila, 1969; Wermelinger and Seifert, 1998, 1999). Besides temperatures, windstorms are importantly interacting with bark beetle disturbances (e.g., Temperli et al., 2013; Thom et al., 2013; Stadelmann et al., 2014), and this interaction can be amplified by climate change (e.g., Seidl and Rammer, 2017). Moreover, windstorms coupled with bark beetle outbreaks remain the most damaging agents in Norway spruce stands as shown, for example, in Slovakia (Kunca et al., 2019). Acute droughts appear to be another important driver of bark beetle infestation as shown on the example of Austria (Netherer et al., 2019) and the influence of droughts on bark beetle outbreaks will even threaten the persistence of European coniferous forests (Jaime et al., 2022). Droughts limiting soil water content contribute to tree transpiration deficit, increasing host susceptibility to bark beetle attacks (Matthews et al., 2018). Effects of summer temperatures, droughts and windstorms on the dynamics of bark beetle outbreaks in Norway spruce forests were analysed across eight European countries by Marini et al. (2017). Great research attention concentrated also on effects of future climate change, represented by various climate scenarios, on different aspects of bark beetle occurrences focusing, for example, on Sweden (Jönsson et al., 2009), Scandinavia (Jönsson et al., 2011), Switzerland (Jakoby et al., 2019), or the Bavarian Forest National Park in Germany (Sommerfeld et al., 2020)."

L66: Here, it is better to start with "In the Czech forestry literature, ..." as this sentence now focusses on Czech republic.
RESPONSE: The recent version of the sentence is: "Information about past Czech bark beetle occurrences was a part of forestry literature (e.g. Chadt-Ševětínský, 1913; Nožička, 1957; Hošek, 1981)."

L66-79: There is a long history of bark beetle impact in forest in the Czech Republic, with focal areas of infestations (Sumava and others), with some exceptional years; this should be said at the end in order to change to the central part of this study: to contextualize the most recent calamity.
RESPONSE: We changed a corresponding part of the text as follows: "Information about past Czech bark beetle occurrences was a part of forestry literature (e.g. Chadt-Ševětínský, 1913;

Nožička, 1957; Hošek, 1981). Some papers concentrated on focal areas of bark beetle outbreaks like the Šumava Mts. after the disastrous windstorm of 26–27 October 1870 (Záloha, 1970) and during the 1870s (Jelínek, 1988), while Zatloukal (1998) analysed factors of past and present bark beetle calamities. PHENIPS model was used to analyse the influence of future climate change on the country bark beetle distribution (Hlásny et al., 2011) and bark beetle dynamic in the Bohemian Forest (Berec et al., 2013). Lubojacký (2012) described the Czech legislation related to protection against bark beetle. Zahradník and Zahradníková (2019) assessed salvage felling caused by bark beetle and other abiotic/biotic factors for 1998–2017. Past and recent bark beetle outbreaks in the Czech Republic after 1980 CE were described, for example, by Skuhravý and Šrot (1988), Mrkva (1993), Skuhravý (2002) and Hlásny et al. (2021b)."

Concentration of the article on "the most recent calamity" follows from the subsequent paragraph.

L81-83: Questions 1 and 2 could be reordered: first the question is about the context of the most recent calamity, and then the question would logically be whether the changing climate has altered the regimes.

RESPONSE: Accepted, we changed it as follows: "Two questions arise. First, how does this outbreak appear in the context of preceding bark beetle outbreaks? And second, does recent climate change completely alter the conditions of such events?"

L86: "analysis and discussion"
RESPONSE: Accepted and corrected.

L86: …outbreaks. Results are discussed in the context of climate change.
RESPONSE: Accepted and corrected.

L87-91: can be deleted.
RESPONSE: Accepted and corrected.

L93: Follow normal paper structure: Intro, Data and methods, Results, Discussion, Conclusion
RESPONSE: Yes, we follow the normal paper structure, dividing Data and methods into separate sections on Data and Methods, as is usual in Climate of the Past.

L94: Section 2.1 could be reduced by putting the data into a table.
RESPONSE: Because Section 2.1 describes the basic bark beetle data, a table would probably not be good solution (moreover, without important reduction of space). We would like to show readers on which types of documentary data (with examples) is our database built. For many readers, documentary data are not as familiar as other ones (such as instrumental records).

L94: According to Wiki: Czech lands, not Czech Lands
RESPONSE: The writing Czech Lands is correct (see use these terms in many our preceding articles)

L130: spatial? if yes: grid data or station data?
RESPONSE: Series are calculated from stations, not gridded data. The construction of corresponding series is in detail described in related papers cited for each of series used.

L146: replace "territorial" by "spatial"
RESPONSE: Accepted and corrected: "The created new database was used to describe long-term spatiotemporal variability of bark beetle occurrences across the CR between 1781 and 1963 CE, from which the three most outstanding bark beetle outbreaks based on the spatial extent and estimated volume of damaged wood were selected for detailed analyses."

L147: replace "for their detailed study" by "for detailed analyses"
RESPONSE: Accepted and corrected, see preceding point.

L 154: replace the beginning of the sentence by "This reference was preferred over the more recent one of 1991-2020 because..."
RESPONSE: The sentence was corrected according to proposal of the native reviewer (N. Macdonald) as follows: "This reference period was preferred to the more recent 1991–2020 period, because of …"

L155: Whatever the intention (more stable or not), the older period was also cooler and more moist.
RESPONSE: This remark is correct only for temperatures, but not for precipitation (see Brázdil et al., 2022b). But to take your comment into account, the related sentence was changed as: "This reference period was preferred to the more recent 1991–2020 period, because of more stable climatic patterns and weaker effect of recent warming (Brázdil et al., 2022b).

L153-164: It is difficult to follow how significance of the anomalies were indeed defined. Some more detail may be helpful.
RESPONSE: Accepted, we add the following paragraph for explanation:
"The method uses a composite matrix in which the rows represent the years of the bark beetle outbreak (event years) and the columns are analysed data (temperature, precipitation, scPDSI) for five years before and after the event (lag years). In random bootstrapping approach, 500 unique versions of composite matrices were created drawing unique subsets of event years at random without replacement. For each column (that is for years from lag–5 to lag+5) and for each normalized composite matrix the density function and its percentiles were calculated. For each lag year the mean data anomaly (temperature, precipitation, scPDSI), that exceeded the 95th percentile, was considered statistically significant (see, e.g., Adams et al., 2003 or Rao et al., 2019 for more details)."
New reference:
Adams, J. B., Mann, M. E., Ammann, C. M.: Proxy evidence for an El Niño-like response to volcanic forcing. Nature, 426, 274, https://doi.org/10.1038/nature02101, 2003.

L180ff: Sections 4.1.1 to 4.1.4 are boring to read and lack reduction. The information could be summarized and the text could go into supplementary material
RESPONSE: We tried to describe it in summarising form giving the most important regional information. In total, it comprises 2 pages of regional information, which we see as quite important to show the spatiotemporal context of bark beetle outbreaks in the Czech Republic. We do not consider removing this information into supplementary material a useful solution.

L264-247: Text can be deleted here.
RESPONSE: It is included here as a logical transition from characteristics of long-term climatic patterns in the context of past 220 years to detail analyses of meteorological and

climatological triggers of selected notable bark beetle outbreaks. It seems, it would be better to preserve it here.

L267-419: I wonder whether it wouldn't be possible to roughly quantify the bark beetle damage in the years before 1980. There is some information mentioned in the text. The text may be too detailed.
RESPONSE: Quantification of the bark beetle damage in the years before 1980 is referred to published information in forestry sources where available. Concerning of three notable bark beetle outbreaks before 1980, no overall country estimation and only selected damage examples exist for the events from the 1830s (Section 4.2.1) and the 1870s (Section 4.2.2); between c. 8 and 10 million m$^3$ of infested wood are existing estimates for the 1940s–1950s (Section 4.2.3).

L452-454: Please clarify the sentence by putting the verb "appear" e.g. after "directions".
RESPONSE: Accepted and corrected.

L573-574: That's exactly the reason for the huge bark beetle calamity: the interaction with windthrow. I bet that salvage logging after windthrow was also the highest ever in CR history. This should be discussed, too.
RESPONSE: Accepted, new Figure 13 comparing bark beetle outbreaks with windstorms was added to Section 5.3 and the corresponding paragraph was changed and complemented as follows: "The recent bark beetle devastation of forests in the CR destroyed in seven years from 2015 to 2021 55.46 mil. m$^3$ of wood altogether, which is more than the total for the previous 51 years (34.80 mil. m$^3$) for which quantitative bark beetle salvage felling data are available (cf. Fig. 2). The highest annual bark beetle felling in 2020 (14.89 million m$^3$) was not exceeded by any other biotic factors (other insects, gnawing of trees by wild animals, mushroom pathogens) nor by any abiotic factors (wind, snow, rime, drought, air pollution). As follows from Figure 13a, bark beetle salvage felling was higher than those caused by windstorms since 1964 only in 1995 and then from 2016 onwards, giving for the last outbreak in 2015–2021 more than three times higher volumes than in case of windstorms (17.03 million m$^3$). The highest windstorm salvage felling was recorded in 1990 with 8.77 million m$^3$ of wood (i.e., before 1992–1995 bark beetle outbreak) and in 2007 with 8.84 million m$^3$, followed in 2008 by 4.85 million m$^3$ (i.e. during 2007–2010 outbreak). Another well-expressed maximum appeared in 1982–1987 outbreak with 5.94 million m$^3$ in 1984 and 6.50 million m$^3$ in 1985. There is a clear relationship to extreme windstorms, such as Kyrill in 2007 or Emma in 2008, while annual frequency of selected severe windstorms across the CR in months January–March together with October–December (Figure 13b) do not show any clear relationship to damaged and processed wood. The dominant proportion of windstorms of the winter half-year on volumes of salvage felling was disturbed only by summer windstorm on 12–13 July 1984 (see Brázdil et al., 2018b)."

[Figure]

Figure 13: Comparison of bark beetle outbreaks and windstorms in the Czech Republic during the 1964–2021 period (NBBO – notable bark beetle outbreak): (a) volumes of annual salvage felling attributed to windstorms (1) and bark beetle infestation (2); (b) annual frequency of severe windstorms for January–March and October–December.

L600-605: This is a poor conclusion. Climate change is not the only reason why bark beetle outbreaks have increased. Rather, there is an interaction of long-term Norway spruce plantations, windthrow events and amplifying effects of warmer temperatures. I wonder to which extent CR would naturally be covered by Norway spruce and what would result as the plantation legacy of the past 200 years. To conclude would mean to promote more deciduous, stand-forming tree species to be planted in the future.

RESPONSE: This paper is oriented on meteorological and climatological triggers of notable bark beetle outbreaks, i.e. our conclusions concentrate particularly on these effects. To solve the whole complex of factors contributing to these events is out of the scope and aims of the article. Our conclusions on L600-605 are following:

"(iv) The last bark beetle outbreak in the CR from the mid-2010s forced by high temperatures and drought has no documented 600 historical analogue to its scope and its devastating effects on Czech spruce forests. It has contributed to the fact that Czech forests have become from 2018 sources of $CO_2$ increase, further increasing the "anthropogenic" footprint of GHG emissions.

(v) Despite great attention devoted to the recent and potential future bark beetle outbreaks forced by recent climate change, there is a great potential to learn more from the past. This study demonstrates a high potential of documentary evidence for the creation of similar databases for other European regions to better understand the natural and socio-economic development that led to the recent situation."

We apologise, but we do not know, what is here wrong. We are not saying here, that climate change is "the only reason why bark beetle outbreaks have increased" (see also Section 5.3).

L940: A figure caption should be self-explaining. Here, a minimum of information on the calculation of the significance should be added.

RESPONSE: Accepted. For better explanation we extended the figure caption as follows:

"Values are anomalies with respect to 1961–1990. Significant anomalies ($p < 0.05$) are indicated with asterisks and they were defined as values exceeding 95 percentiles derived from 500 random samples using bootstrapping method."

L945: The period in the caption in unclear: according to the text, the data refers to four selected outbreaks in the 1980, 1990, 2000, and 2010s; this should be clarified. In addition: why are these outbreaks not addressed in detail? Are they really evenly distributed among the four decades?

RESPONSE: The figure caption was complemented as follows: "Differences in mean relative frequencies of individual circulation types of the objective classification in a composite of summers related to four bark beetle outbreaks from the 1980s (1982–1987, 1992–1995, 2007–2010, 2015–2021), and mean relative frequencies of the 1961–1990 reference period in the Czech Republic …"

The reason for the aggregation of multiple outbreaks for the purpose of circulation analysis is our intention to look for a general pattern that could be attributed to bark beetle outbreaks with lower influence of natural variability than in the case of the analysis of individual years.

---

## Author Comment (AC2)

**Responses to the referee 2 (Neil Macdonald)**

**Overall:** I enjoyed reading this paper this interesting paper that uses a range of archival and instrumental data to better understand the inter-relationship between bark beetle outbreaks and climatic parameters, with considerable impact to the economy of such outbreaks. The wealth of materials that the authors have processed is commendable, with a large volume of materials reviewed and a complex picture starts to emerge. I think that there is considerable merit within the paper, but I do have some concerns with how the paper is currently presented, these are listed below, I also provide an annotated copy to support the authors in the revision process.
RESPONSE: We would like to thank the referee 2 for careful evaluation of our manuscript (including valuable annotated copy) and many critical comments we are trying to respond below.

My key concern is that the paper presents a very detailed understanding of human-environment-climate interactions over several hundred years, with examples recognising these complex interactions, however the conclusion states that this is a function of climate change. Such a statement I feel fails to give justice to the complexity of the different aspects you so carefully detail in the paper. Climate change may have exacerbated the impacts, but I believe you demonstrate a much complex picture of interactions throughout the paper.
RESPONSE: We agree that the problem of bark beetle outbreaks represents complicated "human-environment-climate interactions". But the aim of this paper is to analyse meteorological and climatological triggers of "notable" bark beetle outbreaks. Despite human and environmental factors being partly indicated in several parts of the manuscript, the complex analysis including human and environmental factors is not the aim of this paper and is out of the scope and possibilities of this paper. For these reasons, our conclusions are primarily oriented on the analysis of meteorological and climatological triggers (see the first sentence in Conclusions: "From a systematic analysis of meteorological and climatological triggers of the notable past and present bark beetle outbreaks in the CR for the 18th–21st centuries, the following conclusions can be summarised:"). To clarify more clearly the aim of the paper, we added following sentences at the end of the past paragraph in Introduction: "In order to fulfil this request, the most comprehensive and unique series of bark beetle outbreaks in the Czech Republic was created for this study and used especially for the analysis of meteorological and climatological triggers accompanying the notable bark beetle outbreaks. Having in mind that these outbreaks are a result of complicated human-environment-climate interactions, this study concentrates only on one part of this interaction."

I think you need to present a chronology of windstorms within the paper, these appear important throughout the record, not just the historical period. This could be presented alongside the bark beetle outbreaks to help the reader see the relationship.
RESPONSE: We added 220-year chronology of severe windstorm as part (d) to Fig 3 with the following description at the end of the first paragraph in Section 4.2: "Periods of relatively higher annual frequency of severe windstorms (Fig. 3d) were identified in the 1820s–1840s, the 1900s–1930s, and mainly in the 1960s–2000s, while they were less frequent in the second half of the 19th century (particularly in the 1850s) and in the 1940s–1950s."

[Figure]

Figure 3: Fluctuations in annual and summer series of (a) mean air temperatures, (b) precipitation totals, (c) scPDSI, and (d) severe windstorms for the territory of the Czech Republic during the 1781–2021 period. The values are smoothed by a 10-year Gaussian filter (bold line) and complemented by horizontal lines corresponding to related means of the 1961–1990 period.

Moreover, we added also new Figure 13 comparing bark beetle outbreaks with windstorms to Section 5.3 with changed and complemented text (see below).

**Key points that I believe should be addressed following revision:**
This is a long paper with a great level of detail, whilst this is valuable and informative it does mean you need to ensure that key points are clearly detailed. I have suggested a couple of places where this could be more concise in the annotated version.
RESPONSE: We accepted your proposed corrections from the annotated version.

Lines 50-80, I think this needs to be more concise, it is not incorrect but not critical to the readers understanding.
RESPONSE: Following your request, we changed related two paragraphs from lines 50-80 as follows:
"Among different aspects of bark beetle outbreaks, several papers analysed in great detail the effect of temperatures on the duration of the egg, larval, and pupal stages of the spruce bark beetle and adult maturation feeding (e.g., Annila, 1969; Wermelinger and Seifert, 1998, 1999). Besides temperatures, windstorms are importantly interacting with bark beetle disturbances (e.g., Temperli et al., 2013; Thom et al., 2013; Stadelmann et al., 2014), and this interaction can be amplified by climate change (e.g., Seidl and Rammer, 2017). Moreover, windstorms coupled with bark beetle outbreaks remain the most damaging agents in Norway spruce stands as shown, for example, in Slovakia (Kunca et al., 2019). Acute droughts appear

to be another important driver of bark beetle infestation as shown on the example of Austria (Netherer et al., 2019) and the influence of droughts on bark beetle outbreaks will even threaten the persistence of European coniferous forests (Jaime et al., 2022). Droughts limiting soil water content contribute to tree transpiration deficit, increasing host susceptibility to bark beetle attacks (Matthews et al., 2018). Effects of summer temperatures, droughts and windstorms on the dynamics of bark beetle outbreaks in Norway spruce forests were analysed across eight European countries by Marini et al. (2017). Great research attention concentrated also on effects of future climate change, represented by various climate scenarios, on different aspects of bark beetle occurrences focusing, for example, on Sweden (Jönsson et al., 2009), Scandinavia (Jönsson et al., 2011), Switzerland (Jakoby et al., 2019), or the Bavarian Forest National Park in Germany (Sommerfeld et al., 2020).

Information about past Czech bark beetle occurrences was a part of forestry literature (e.g. Chadt-Ševětínský, 1913; Nožička, 1957; Hošek, 1981). Some papers concentrated on focal areas of bark beetle outbreaks like the Šumava Mts. after the disastrous windstorm of 26–27 October 1870 (Záloha, 1970) and during the 1870s (Jelínek, 1988), while Zatloukal (1998) analysed factors of past and present bark beetle calamities. PHENIPS model was used to analyse the influence of future climate change on the country bark beetle distribution (Hlásny et al., 2011) and bark beetle dynamic in the Bohemian Forest (Berec et al., 2013). Lubojacký (2012) described the Czech legislation related to protection against bark beetle. Zahradník and Zahradníková (2019) assessed salvage felling caused by bark beetle and other abiotic/biotic factors for 1998–2017. Past and recent bark beetle outbreaks in the Czech Republic after 1980 CE were described, for example, by Skuhravý and Šrot (1988), Mrkva (1993), Skuhravý (2002) and Hlásny et al. (2021b)."

155-157, I am uncomfortable with the way you are using the scPDSI, why not just present this as scPDSI-12?
RESPONSE: According to our opinion, individual notable bark beetle outbreak are results of some cumulative effects. Because we are presenting the cumulative effects of monthly temperatures and precipitation, we logically collected the same monthly data for PDSI. But based on your comment, in part (b) of Figures 4–10 we replaced all existing annual data by scPDSI-12 (and scPDSI-3 for summer).

470-480 I felt that you needed to explain the spatial and temporal changes in land management earlier, could you present changing extent of forestry in CR through a table or graph underneath Fig 1 or 2. I felt you also need to discuss the different forestry practices of the times, you recognise that extensive lowland forestry took place in the mid-20C but fail to then consider has the increase you attribute to climate change just been a function of these woods maturing. Also what impact does large monoculture conifer plantations have, compared to mixed conifer/deciduous?
RESPONSE: As mentioned earlier, this paper has not ambition for the complex analysis of complicated human-environment-climate interactions related to bark beetle outbreaks. Our aim is to characterise only meteorological and climatological triggers of notable bark beetle outbreaks with respect to their changes during past 220 years, i.e. were they the same during this long period or we may see there some imprints of recent climate change connected with increasing temperatures and dryness? As for monocultures and forestry practices in general, these are dealt with in Section 5.3.

480 Do we get different types of bark beetle, are they responsible for different outbreaks, do you need to try and distinguish between different beetle outbreaks and the different climates they preference, do we have different beetles impacting forests in the last 50 years?

RESPONSE: Our study concentrates primarily on *Ips typographus* while information of other bark beetles (*Ips duplicatus*; *Pityogenes chalcographus*) was not considered in this paper. As follows from volumes of salvage felling related to other bark beetles, their total volumes of infested wood are negligible compared to *Ips typographus* (only *c*. 0.1%).

545 A little more detail needed.

RESPONSE: It was extended as follows: "The planting of extensive conifers monocultures reduced the biodiversity of Czech forests, exhausted the soil and increased the susceptibility of forests to natural disasters (e.g., through even-aged structure, mutual competition for essential nutrients, lack of space for the root system and treetop) (Daniel et al., 2013). Without the existence of conifers monocultures in the lowlands and at middle altitudes, forests would be composed especially of oak, beech and fir (Neuhäuslová et al., 1997). However, in the last few years, the area covered by spruce monoculture plantations in the CR declined from 54.1 % in 2000 to about 48.8 % in 2020, which should contribute to the attenuation of the above-mentioned negative phenomena (Ministerstvo zemědělství, 2021)."

New reference:

Neuhäuslová, Z., Moravec, J., Chytrý, M., Sádlo, J., Rybníček, K., Kolbek, J., Jirásek, J.: Map of potential natural vegetation of the Czech Republic 1:500,000, Botanický ústav AV ČR, Průhonice, 1997.

Discussion: You need to provide a more detailed analysis of the role of windstorms, these appear to be significant throughout the record, not just historically, this needs to be explored more fully.

RESPONSE: We newly added Figure 13 comparing bark beetle outbreaks with windstorms to Section 5.3 and the corresponding paragraph was changed and complemented as follows: "The recent bark beetle devastation of forests in the CR destroyed in seven years from 2015 to 2021 55.46 mil. m$^3$ of wood altogether, which is more than the total for the previous 51 years (34.80 mil. m$^3$) for which quantitative bark beetle salvage felling data are available (cf. Fig. 2). The highest annual bark beetle felling in 2020 (14.89 million m$^3$) was not exceeded by any other biotic factors (other insects, gnawing of trees by wild animals, mushroom pathogens) nor by any abiotic factors (wind, snow, rime, drought, air pollution). As follows from Figure 13a, bark beetle salvage felling was higher than those caused by windstorms since 1964 only in 1995 and then from 2016 onwards, giving for the last outbreak in 2015–2021 more than three times higher volumes than in case of windstorms (17.03 million m$^3$). The highest windstorm salvage felling was recorded in 1990 with 8.77 million m$^3$ of wood (i.e., before 1992–1995 bark beetle outbreak) and in 2007 with 8.84 million m$^3$, followed in 2008 by 4.85 million m$^3$ (i.e. during 2007–2010 outbreak). Another well-expressed maximum appeared in 1982–1987 outbreak with 5.94 million m$^3$ in 1984 and 6.50 million m$^3$ in 1985. There is a clear relationship to extreme windstorms, such as Kyrill in 2007 or Emma in 2008, while annual frequency of selected severe windstorms across the CR in months January–March together with October–December (Figure 13b) do not show any clear relationship to damaged and processed wood. The dominant proportion of windstorms of the winter half-year on volumes of salvage felling was disturbed only by summer windstorm on 12–13 July 1984 (see Brázdil et al., 2018b)."

[Figure]

Figure 13: Comparison of bark beetle outbreaks and windstorms in the Czech Republic during the 1964–2021 period (NBBO – notable bark beetle outbreak): (a) volumes of annual salvage felling attributed to windstorms (1) and bark beetle infestation (2); (b) annual frequency of severe windstorms for January–March and October–December.

Conclusions: I think the paper demonstrates a much more complex picture that cannot simply be brought down to the points presented here. Looking a the data I think you need to reflect on iii) as I do not see this clearly within the data. You provide a wealth of information in the paper detailing the complex interactions that may explain changing patterns of forestry, landuse patterns, management practices, beetle outbreaks and climatology, all of which contribute. To argue recent outbreaks are a function of climate change fails to acknowledge the evidence presented in the paper, that demonstrates an emerging complex picture, of which climate change may be exacerbating the incidences, which are a function of complex human-environment interactions.

RESPONSE: We agree that the problem of bark beetle outbreaks represents complex human-environment-climate interactions. But the aim of this paper is to analyse meteorological and climatological triggers of "notable" bark beetle outbreaks. Despite human and environmental factors being partly indicated in several parts of the manuscript (just to show that not only climate is important), the complex analysis including human and environmental factors is not the aim of this paper and it is out of the scope and possibilities of this paper as mentioned in the last paragraph of Introduction. For these reasons, our conclusions are primarily oriented on the analysis of meteorological and climatological triggers (see also the first sentence in Conclusions: "From a systematic analysis of meteorological and climatological triggers of the notable past and present bark beetle outbreaks in the CR for the 18th–21st centuries, the following conclusions can be summarised:"). We hope, that our short conclusions reflect representatively our main meteorological and climatological findings related to available data.

Figure 1&2: It is crucial to present the windstorm data alongside this information. You present a strong argument for windthrow damage in the earlier period, but how does this coincide with the bark beetles, without a long chronology of such information it is difficult for the reader to discern what role this has, and whether it is diminished in significance in the last 50 years.
RESPONSE: To demonstrate fluctuations of windstorms in the whole period, we added series of severe windstorms for the Czech Republic from Brázdil et al. (2018b) extended on the

whole period analysed (Figure 3d). Because our study concentrates particularly on seven notable bark beetle outbreaks, corresponding extreme windstorms are always reported in these individual cases. Relationships about windstorms and bark beetle outbreaks are reported also in Section 5.3 with new Figure 13.

Figure 4-6: Why not present a scPDSI-12, -18 or -24, this would be better than a cumulative monthly (scPDSI-1?) anomaly of the scPDSI for the timescales presented. It would be helpful for the reader to see the bark beetle patterns of damage alongside the climatological information, add graph below scPDSI annual/summer anomaly.
RESPONSE: According to our opinion, individual notable bark beetle outbreak are results of some cumulative effects. Because we are presenting the cumulative effects of temperatures and precipitation in monthly data, we logically collected the same monthly data for PDSI. But based on this comment, in part (b) of Figures 4–10 we replaced existing data by scPDSI-12. In case of the use of scPDSI-18 or -24 we are not sure, if it is not a mixture of two vegetation seasons together (but there is not any problem to mention them in a supplementary material).

Please see annotated paper for suggested minor modifications
RESPONSE: We corrected all particular parts of text and sentences as you proposed a considered your further comments in annotated text for further corrections.